# Recognize Your Orchestrator: An Entropy Dynamics Perspective for LLM Multi-Agent Systems

**Junze Zhu** [1 2]  **Weihao Chen** [1 2]  **Xuanwang Zhang** [1 2]  **Zhen Wu** [1 2]  **Xinyu Dai** [1 2]

## Abstract

The transition from single-turn models to Multi-Agent Systems (MAS) promises enhanced problem-solving capabilities, yet the centralized orchestration topology remains a critical point of fragility. To analyze this, we propose a Mean-Field Entropy Dynamics framework, modeling the orchestration process as a system governed by the competing forces of task resolution and cumulative context loading. To facilitate validation, we introduce Inverse Workflow Generation (IWG), a multi-agent pipeline that synthesizes process-verifiable, high-complexity benchmarks with dense intermediate checkpoints. We demonstrate that our entropy dynamics model fits empirical trajectories, providing physically interpretable parameters that quantify system stability and performance collapse. Crucially, our analysis uncovers a "Reasoning Trap": while reasoning-heavy models excel in isolated tasks, they frequently fail as orchestrators due to context squeezing. Elucidating the physical mechanisms underlying the Orchestrator and quantifying systemic uncertainty offers insights for the MASs' architectural design. Our code is available at https://github.com/NJUNLP/orchestrator_entropy.

## 1. Introduction

The paradigm of Artificial Intelligence has shifted rapidly from single-turn chatbots to autonomous agents capable of tool use and multi-step reasoning (Wei et al., 2022; Yao et al., 2023b). While agents operating under frameworks like ReAct (Yao et al., 2023b) have demonstrated proficiency in isolated tasks, they often struggle with long-horizon collaborative tasks due to limited context windows and the accumulation of reasoning errors (Shinn et al., 2023).

To mitigate these limitations, multi-agent systems (MAS) have been proposed as a robust solution. By distributing cognitive load across specialized roles—such as Coders, Researchers, and Reviewers—MAS aims to emulate human organizational structures. Among the prevailing architectures, the Centralized Orchestration paradigm (or Hierarchical/Manager-Executor paradigm) has become the standard for general-purpose applications (Wu et al., 2023; Hong et al., 2023). In this topology, a high-level Orchestrator agent functions as the central executive, responsible for breaking down the original query into atomic plans and scheduling execution tasks for available worker agents.

However, the efficacy of this centralized control remains a subject of debate. While recent frameworks like Magentic-One (Research, 2024) and WebPilot (Zhang et al., 2025b) achieve superior performance through functional specialization and centralized management, their complex collaborative mechanisms inherently introduce system vulnerability. We argue that the vulnerability stems primarily from the collapse of the Orchestrator, making it the bottleneck of the system. To diagnose the root causes of fragility in current Multi-Agent Systems (MAS), we conducted a comprehensive failure attribution analysis across four representative domains (DeepResearch (Shao et al., 2024), AgentCoder (Hong et al., 2023), GUIBrowser (Erdogan et al.), AgenticRAG (Papageorgiou et al., 2025)) on various mainstream benchmarks. Our empirical findings, detailed in Appendix A, reveal a critical insight: the Orchestrator is responsible for a substantial majority of all task failures, whereas individual Executors account for only 32.4%.

The above "Orchestrator Bottleneck" suggests that failure typically arises not from a lack of tool capabilities, but from the inability to maintain global logical oversight and effectively manage the inherent uncertainties during long-horizon planning. This empirical disparity underscores a fundamental gap in our understanding of how Orchestrators manage complex decision-making processes. However, existing empirical-based evaluations and analysis are insufficient to capture the nuanced interplay between task complexity and system-wide uncertainty.

[1]National Key Laboratory for Novel Software Technology, Nanjing University, China [2]School of Artificial Intelligence, Nanjing University, China. Correspondence to: Zhen Wu <wuz@nju.edu.cn>.

*Proceedings of the 43ʳᵈ International Conference on Machine Learning*, Seoul, South Korea. PMLR 306, 2026. Copyright 2026 by the author(s).

To bridge this gap, it is imperative to formalize the underlying mechanism of orchestration failures through a rigorous mathematical lens. Based on this, we propose a rigorous theoretical and empirical framework to deconstruct the dynamics of multi-agent orchestration. First, we establish a Mean-Field Entropy Dynamics model, deriving a deterministic dynamics equation that links microscopic scheduling decisions to the macroscopic evolution of system entropy, characterized by transient task resolution and cumulative context loading. Furthermore, to support Mean-Field Entropy Dynamics modeling, we introduce the Inverse Workflow Generation (IWG) pipeline, which leverages an inverse planning mechanism to synthesize intermediate processes verifiable task benchmarks. Finally, we conduct a comprehensive evaluation across proprietary and open-source models, and validate that our entropy model parameters serve as robust indicators for the cognitive ability and stability of LLM-based Orchestrators.

## 2. Mean-Field Entropy Dynamics Modeling

This section establishes a theoretical framework that bridges the discrete, microscopic scheduling decisions in a multi-agent system with the continuous, macroscopic evolution of scheduling entropy. We begin with a formal description of the discrete decision process, yet crucially, we reinterpret the sequence of scheduling vectors as stroboscopic observations of a latent continuous probability field. Drawing on the theoretical insight that sampling dynamics can be rigorously modeled as gradient flows in the space of probability measures (Wibisono, 2018), we derive a mean-field approximation (Lasry & Lions, 2007) for the evolution of the expected scheduling entropy.

### 2.1. Formal Description of the Scheduling Process

Consider an LLM-based multi-agent system consisting of $n$ agents responsible for specific tasks, denoted as Executors $\mathcal{E} = \{e_1, e_2, \ldots, e_n\}$, operating under the control of an Orchestrator $O$. At each discrete time step $k = 0, 1, 2, \ldots$, the system maintains a *global context* $\mathcal{C}_k$. This context is initialized with the user's original query $Q$ (i.e., $\mathcal{C}_0 = \{Q\}$) and is iteratively updated by appending the response outputs of selected executors. The Orchestrator functions as a policy network that maps the current context $\mathcal{C}_{k-1}$ to a decision space over $\mathcal{E}$. Specifically, at step $k$, the Orchestrator generates a scheduling probability vector $\mathbf{p}_k = (p_1(k), p_2(k), \ldots, p_n(k))^\top$, where each scalar component $p_i(k) = \mathcal{P}(e_k = e_i \mid \mathcal{C}_{k-1})$ represents the conditional probability that executor $e_i$ is the optimal candidate to handle the current state. This vector quantifies the relevance of each agent given the accumulated history, satisfying $\sum_{i=1}^n p_i(k) = 1$. The executor $e_k$ is then sampled or selected based on $\mathbf{p}_k$, and its output is appended to

*Table 1.* Nomenclature and Definitions of System Parameters.

| Symbol | Definition |
|---|---|
| $\mathcal{E}$ | Set of executor agents, $\mathcal{E} = \{e_1, \ldots, e_n\}$. |
| $O$ | The Orchestrator agent responsible for scheduling. |
| $k$ | Discrete time step index ($k \in \mathbb{N}$). |
| $\mathbf{p}_k$ | Scheduling probability vector at step $k$, $\sum p_i(k) = 1$. |
| $\mathcal{C}_k$ | Global context at step $k$, comprising the query $Q$ and sequence of past agent outputs. |
| $\bar{H}(t)$ | Macroscopic expected scheduling entropy. |

form $\mathcal{C}_k$. Key notations are summarized in Table 1.

Orchestrator handles user requests by disassembling tasks and scheduling executors in a certain multi-agent system. Therefore, in order to explore the ability of LLM as orchestrator, we need to model the whole multi-agent system. Modeling the complete $n$-agent system poses a fundamental challenge due to the high-dimensional, stochastic interactions among agents mediated through the shared context $\mathcal{C}_k$ and the orchestrator's policy.

### 2.2. Mean-Field Approximation of the Orchestrator's Scheduling Policy

Modeling the exact microscopic interactions in a multi-agent system is computationally intractable due to the high-dimensional, stochastic nature of the Orchestrator's decision-making process. To obtain a tractable description, we employ a **mean-field approximation** interpreted through a statistical ensemble perspective, drawing inspiration from (Mi et al., 2025; Huang et al., 2007).

Specifically, we model the Orchestrator's scheduling policy as a **probability field** at continuous time $t$ evolving over the discrete space of executors. In this framework, the complex history-dependent interactions are decoupled: the Orchestrator acts as the generator of a self-consistent mean field, represented by the scheduling probability vector $\mathbf{p}_t$. Each executor selection event is thus a realization drawn from this mean field. To embed discrete scheduling orchestration into a latent continuous probability space, we adopt a field-theoretic perspective in which the discrete scheduling vector $\mathbf{p}_k$ is a sampled observation from an underlying continuous probability density evolving over the executor space. Formally, we treat the Orchestrator's policy as a time-dependent probability measure $\Psi_t$ on $\mathcal{E}$, whose dynamics are driven by the accumulated context $\mathcal{C}_t$. The executor selection process corresponds to the collapse of the wave function into a specific agent state.

To quantify the dynamics of the Orchestrator, we utilize the scheduling entropy as the order parameter (Macroscopic Indicator), aligning with maximum entropy principles in

*Table 2.* Physical Interpretation of Model Parameters: Mathematical Meanings and System Implications.

| Parameter | Mathematical Meaning | Physical Implication |
|---|---|---|
| $A_{\text{task}}$ | **Initial Amplitude**: Determines the magnitude of the starting entropy spike and subsequent oscillations. | **Task Complexity**: Represents the initial ambiguity or difficulty of the user query. Higher complexity leads to broader initial agent exploration. |
| $\gamma$ | **Damping Coefficient**: Controls the rate at which the oscillation envelope decays. | **Solving Efficiency**: Reflects the system's ability to converge on a solution. A higher $\gamma$ implies faster agent consensus and task resolution. |
| $\omega$ | **Angular Frequency**: Determines the density of peaks and troughs within a time period. | **Exploration Granularity**: Represents the pace of the explore-exploit cycle. Higher frequency suggests rapid switching between sub-tasks or trial-and-error attempts. |
| $\phi$ | **Initial Phase**: The initial Phase at $t = 0$ determines the initial state of motion. | **Prompt Alignment Bias**: The initial phase of Orchestrator is instantly triggered solely by the semantics of the user query. |
| $\beta$ | **Logarithmic Slope**: Coefficients scaling the persistent growth term $(\ln t)$. | **Context Sensitivity**: Measures how severely the accumulation of history creates cognitive load, hindering the orchestrator's focus. |
| $H_0$ | **Vertical Intercept**: The entropy value of the baseline offset. | **Intrinsic Uncertainty**: The irreducible stochasticity of the LLM itself, independent of the specific task or context history. |

decision processes (Ziebart et al., 2008), which serves as a measure of the wave packet's delocalization. A high entropy implies a diffuse wave packet (uncertainty in agent selection). The expected entropy $\bar{H}(t)$ is defined self-consistently by the distribution of agent selection probabilities:

$$\bar{H}(t) = -\sum_{i=1}^{n} p_i(t) \log_2 p_i(t). \quad (1)$$

By tracking $\bar{H}(t)$, we effectively observe the scalar projection of the system's high-dimensional evolution. This approximation allows us to transition from discrete, stochastic agent sampling to a deterministic differential equation describing the fluidity of the reasoning process.

### 2.3. Parameterizing the Entropy Dynamics Equation

To obtain the dynamics of $\bar{H}(t)$, it's necessary to model the evolution of the Orchestrator's probability state $\Psi_t$. We posit that the time derivative of the macroscopic entropy is driven by the superposition of two distinct state update operators acting on the probability field:

$$\frac{d\bar{H}}{dt} = \mathcal{F}_{\text{task}}[\Psi_t] + \mathcal{D}_{\text{context}}[\Psi_t] \quad (2)$$

where $\mathcal{F}_{\text{task}}$ represents the *Focusing Operator* driven by task resolution, and $\mathcal{D}_{\text{context}}$ represents the *Dispersion Operator* driven by context accumulation.

**Focusing Operator: Optimization with Momentum** The primary goal of the Orchestrator is to collapse the probability distribution onto the optimal executor subset. We model this physically as an optimization process minimizing a cognitive potential on the probability manifold.

Crucially, LLM-based Orchestrators do not update policies memorylessly; they possess cognitive inertia derived from the attention mechanism over historical tokens. In optimization theory (Su et al., 2016), this inertia is analogous to

momentum. The continuous limit of such momentum-based optimization behaves, a phenomenon termed hunting behavior, as a **damped harmonic oscillator**. Consequently, the entropy reduction follows the envelope of this underdamped relaxation:

$$\mathcal{F}_{\text{task}} \rightarrow \frac{d}{dt}\left(A_{\text{task}} e^{-\gamma t} \sin(\omega t + \phi)\right) \quad (3)$$

**Dispersion Operator: Context-Induced Diffusion** Simultaneously, as the context window grows, the dimensionality and noise of the input space increase, the continuous accumulation of the global context $\mathcal{C}_t$ exerts an expansive pressure on Orchestrator. We model this as a **diffusion process**. In a standard diffusion regime, the variance of a wave packet spreads linearly with time due to fluctuations. Since the differential entropy of a distribution scales with the logarithm of its variance, the rate of entropy production due to context loading scales inversely with time:

$$\mathcal{D}_{\text{context}} \rightarrow \frac{\beta}{t+1} \quad (4)$$

**Unified Macroscopic Equation** Integrating the contributions from both operators (Eq. 2) yields the final analytical form for the mean scheduling entropy path. This equation unifies the coherent hunting behavior of task resolution with the incoherent dispersion of context loading:

$$\bar{H}(t) = \int_0^t \left(\mathcal{F}_{\text{task}} + \mathcal{D}_{\text{context}}\right) dt \quad (5)$$

$$= A_{\text{task}}\, e^{-\gamma t} \sin(\omega t + \phi) + \beta \ln(t+1) + H_0 \quad (6)$$

The parameters $\Theta = (A_{\text{task}}, \gamma, \omega, \beta, H_0)$ provide a physical link between the macroscopic entropy curve and the microscopic mechanical properties of the Orchestrator, effectively parameterizing the trade-off between reasoning convergence and context management. The physical interpretation of these parameters is detailed in Table 2, with full theoretical derivations provided in Appendix B.

# 3. Inverse Workflow Generation

As established in Section 2, the mean-field entropy dynamics model requires observing the orchestrator's scheduling entropy $\bar{H}(t)$ at every step, and this requirement can only be satisfied by dense step-level execution logs. Yet, standard multi-agent benchmarks, such as GAIA (Mialon et al., 2023), AssistantBench (Yoran et al., 2024), are process-opaque: they provide only the initial query and a final answer check, without the intermediate scheduling states needed to fit the continuous-time dynamics. To bridge this gap, we introduce the Inverse Workflow Generation (IWG) pipeline. Unlike prior data collection strategies that passively record forward executions, IWG adopts an inverse-synthesis paradigm: starting from the target solution, it explicitly reconstructs the necessary environment states and tool outputs that would lead a capable orchestrator to that solution. Crucially, IWG does not directly fabricate agent trajectories; instead, it synthesizes the interactive environment (including tool responses and evidence-bearing observations). The actual orchestration trajectory, comprising the per-step scheduling vectors $\mathbf{p_k}$ and executor selections, is subsequently obtained by running the orchestrator and its executor agents in this synthesized environment. This design guarantees the availability of dense, step-level observation checkpoints, thereby producing the high-resolution data necessary to fit the system dynamics equation while preserving the genuine execution behavior of the models under study. To implement this inverse synthesis, the pipeline employs a collaborative multi-agent workflow shown in Figure 1. The process begins with Scout agent, which recursively deduces the necessary intermediate tasks from the seed data. These task marks are then instantiated by Wrapper agent, which generates the query-response history and environmental information. To guarantee data fidelity, a Validation Committee enforces a three-tier quality check, verifying that the generated path is both logically sound and executable. Detailed workflow diagram and a step-by-step case study are provided in Appendix C.

## 3.1. Scout Agent: Inverse Planning & Decomposition

The Scout Agent serves as the architect of the reverse trajectory. It accepts the **Seed Data** (e.g., QA pair) and the **Target MAS Configuration** (e.g., tool sets, capability description) as inputs. Unlike forward planners that explore potential paths, the Scout operates on capability-aware inverse analysis. It performs a recursive dependency check starting from the final answer. As illustrated in the Figure 1, the Scout initiates the planning from the final answer *March 23, 1942*. It identifies that this date is an attribute of the entity *Michael Haneke*. Consequently, it establishes the task $M_1$ (Identify Director) and maps it to the *Entity Retriever* agent. This process generates a graph of Task Marks ($M_L$), ensuring that every step is both logically necessary for the final an-

swer and strictly executable by the available agents. At this stage, the Scout defines what information type is needed and pass this requirement to the wrapper to materialize the environment content that agents will interact with.

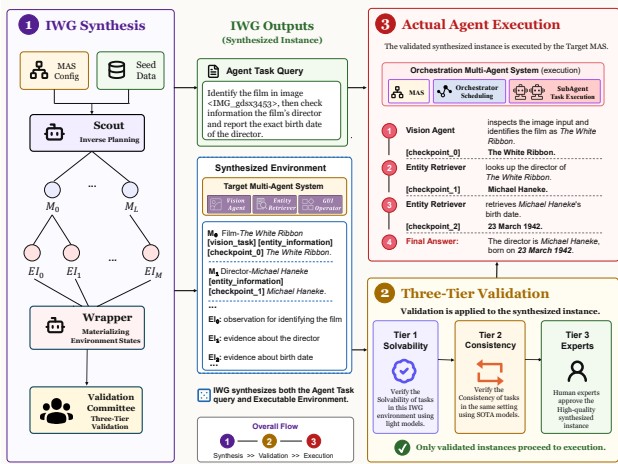

*Figure 1.* System Architecture of Inverse Workflow Generation.

## 3.2. Wrapper Agent: Environment Synthesis

While the Scout defines the logical skeleton, the Wrapper Agent performs **Environment Synthesis**, transmuting abstract milestones into an executable interactive environment. Its primary function is to materialize the key evidence required for agent reasoning, rather than simply supplying answers. Specifically, the Wrapper instantiates each task mark ($M_L$) into a simulated environment state ($EI_M$). As shown in Figure 1, for $M_0$ (Recognize Film), the Wrapper does not output the tag *The White Ribbon* directly; instead, it synthesizes a determined tool outputs ($EI_0$) reflected in the natural language response as the Vision Agent's observation. This provides the necessary context for the subsequent agent to formulate the query about the director. Furthermore, to enforce rigorous evaluation, the Wrapper embeds deterministic **Checkpoints** within these states. Rather than LLM subjective scoring, Wrapper construct the explicit verification logic (e.g., exact-match validating the string *Michael Haneke* at Checkpoint 1). Step-level checkpoints ensure that every intermediate step is factually accurate before the workflow proceeds.

## 3.3. Validation Committee: Quality Control

To strictly align the synthesis with the experimental requirements, i.e., the environment and the action must have significant causation. We implement a **Three-Tier Validation Protocol** to ensure that every Checkpoint acts as a necessary deduction from the synthesized $EI$. **Tier 1 (Solvability Check):** An open-source model acts as the first examiner. The instance is retained only if the model can successfully in-

fer the pre-defined checkpoints using the Wrapper-generated $EI$. **Tier 2 (Consistency Check):** Surviving instances are re-evaluated by a stronger proprietary model. This confirms that the reasoning path is reproducible across different architectures, ensuring the task is universally solvable. **Tier 3 (Expert Review):** The final stage involves Human-in-the-Loop verification to review factual correctness and logical coheren ce. Human Experts Filtering guarantees that the final dataset contains strictly correct task and environment setting.

# 4. Experimental Result and Analysis

## 4.1. Experimental Setting

We instantiated a multi-agent system with $n = 7$ LLM Executor agents and an orchestrator agent $O$. The executors were implemented with the GPT-4 and GPT4-o as the backbone models and ReAct architecture with specialized tool sets. The prompt settings and a complete workflow case of the MAS-Orchestrator and IWG used in the experiment are recorded in the Appendix G. A diverse agent task set of 679 instances from IWG was gathered, spanning domains such as complex data analysis (Chen et al., 2021), research report writing (Yang et al., 2018), omni task (Mishra et al., 2019). For each task $Q_i$, the system was run until task completion or a maximum step limit ($k_{\max} = 20$).

To rigorously evaluate the general capability of LLMs acting as central orchestrators in complex multi-agent environments, we benchmarked a comprehensive suite of models, categorized into Proprietary Models (e.g., Gemini-2.5-Pro, Claude-4.5-Sonnet, GPT-5) and Open-Source Models (e.g., DeepSeek-V3.1, Llama3, Qwen-3). All models were tested under identical prompt constraints to ensure fairness.

We employed a two-tiered evaluation framework to capture both the macro-level task completion and the micro-level orchestration behavior. System-Level metrics assess the overall effectiveness of the multi-agent system driven by the LLM orchestrator, while Orchestrator-Level metrics provide fine-grained insights into the decision-making process. The complete definition and calculation process of these indicators are displayed in the Appendix D.

**System-Level Metrics** *TS (Task Success Rate)* denotes the primary indicator of whether the user's intent was fully achieved. *LCS-F1* is a text-overlap metric based on the Longest Common Subsequence, measuring the structural similarity of Executor Scheduling between the generated solution and the optimal solution.

**Orchestrator-Level Metrics** *Step-SR (Step Success Rate)* measures the accuracy of individual steps within the planning trajectory. *EH-F1 (Exception Handling F1-Score)* evaluates the model's robustness in detecting and recovering

from executor errors. *Consistency* represents the degree of context coupling and logical coherence throughout the interaction. *Faithfulness* assesses the orchestrator's ability to accurately comprehend and utilize the outputs provided by executor agents without hallucination.

## 4.2. Result Analysis of LLM as Orchestrator

The experimental results, as summarized in Table 3 and Tabel 4, reveal significant performance disparities across different model architectures. Rather than treating it as a model leaderboard, we analyze the performance disparities through the lens of entropy dynamics. Based on this observation, we identify two primary archetypes of orchestrator behavior: **Oscillatory-Aggressive** and **Smooth-Conservative**.

**Oscillatory-Aggressive** (e.g., DeepSeek-V3.1, Qwen-3-Max): These models are characterized by high task amplitude ($A_{\text{task}}$) and frequency ($\omega$), indicating broad initial exploration and rapid hypothesis switching. This suggests an aggressive initial decomposition strategy. As shown in Table 3, DeepSeek-V3.1 achieves high structural similarity (LCS-F1: 80.12%) but lower final task success (TS: 29.69%). In our theory, it means that the rapid expansion of the probability wave packet, quantified by high $A_{\text{task}}$ and $\omega$, temporarily exceeds the orchestrator's epistemic threshold.

**Smooth-Conservative** (e.g., Gemini-2.5-Pro, Claude-4.5-Sonnet): These models show a more gradual increase in entropy dominated by context loading rather than task oscillation. The lower $\omega$ and $\beta$ values indicate a steady, deliberate scheduling policy. Claude achieves the highest Step-SR (60.83%) and maintains remarkable consistency (83.52%). This aligns with an overdamped or critically-damped system characterized by low angular frequency $\omega$ and, crucially, low context sensitivity $\beta$. These models sacrifice rapid hypothesis switching for stable, breadth-first planning, effectively suppressing the context dispersion.

Besides, certain models exhibit balanced or hybrid behavioral patterns that resist classification into the two extreme regimes discussed above. Specifically, **Grok-4** and **GPT-4.1** demonstrate robust equilibrium between exploration and stability, achieving respectable Step-SR (42.67% and 36.61%) alongside high Exception Handling F1 (90.61 and 88.57).

Because of the stringent capability requirements of the role of LLM-as-Orchestrator, the majority of models particularly with limited context windows or insufficient instruction-following precision, fail to maintain coherent multi-step delegation, resulting in depressed Task Success rates that render them impractical for complex orchestration.

## 4.3. Entropy Dynamics Validation and Analysis

In this section, we validate the proposed Mean-Field Entropy Dynamics model against empirical data collected from

*Table 3.* Performance Comparison of LLM as Orchestrator

| Model | System-Level | | | Orchestrator-Level | | | | |
|---|---|---|---|---|---|---|---|---|
| | LCS-F1 | TS | Avg | Faithfulness | EH-F1 | Step-SR | Consistency | Avg |
| *Proprietary Models* | | | | | | | | |
| G Gemini-2.5-Pro | 81.28 | 44.14 | 62.71 | 66.19 | 91.72 | 53.33 | 83.46 | 73.68 |
| G Gemini-2.5-Flash | 77.43 | 43.83 | 60.63 | 67.68 | 85.02 | 47.11 | 84.38 | 71.05 |
| Claude-4.5-Sonnet | 85.38 | 29.31 | 57.35 | 66.17 | 97.19 | 60.83 | 83.52 | 77.18 |
| Claude-3.5-haiku | 51.35 | 30.62 | 40.99 | 69.95 | 65.48 | 31.78 | 74.17 | 60.35 |
| Grok-4 | 63.99 | 24.48 | 44.24 | 63.97 | 90.61 | 42.67 | 82.51 | 69.94 |
| Grok-3 | 55.95 | 32.33 | 44.14 | 69.70 | 87.29 | 40.89 | 76.35 | 68.56 |
| GPT-4.1 | 49.96 | 29.55 | 39.76 | 69.91 | 88.57 | 36.61 | 80.28 | 68.84 |
| GPT-5 | 33.27 | 11.71 | 22.49 | 62.51 | 92.65 | 25.11 | 76.45 | 64.18 |
| o4-mini | 32.81 | 10.76 | 21.79 | 69.76 | 76.60 | 26.22 | 76.43 | 62.25 |
| Qwen-Max | 49.96 | 25.36 | 37.66 | 69.72 | 91.62 | 30.16 | 82.89 | 68.60 |
| Qwen-Plus | 36.80 | 12.57 | 24.69 | 68.95 | 82.69 | 28.16 | 82.19 | 65.50 |
| Doubao-1.5-Pro | 42.81 | 13.74 | 28.28 | 68.13 | 82.67 | 34.67 | 79.01 | 66.12 |
| Doubao-Seed-1.6 | 33.71 | 10.88 | 22.30 | 69.68 | 80.71 | 24.00 | 80.19 | 63.65 |
| *Open-Source Models* | | | | | | | | |
| DeepSeek-V3.1 | 80.12 | 29.69 | 54.91 | 66.54 | 74.58 | 42.89 | 76.51 | 65.13 |
| DeepSeek-R1 | 23.54 | 7.43 | 15.49 | 58.22 | 25.35 | 31.56 | 80.72 | 48.96 |
| Llama4-maverick | 67.19 | 36.14 | 51.67 | 67.50 | 76.39 | 39.33 | 79.30 | 65.63 |
| Llama4-scout | 35.13 | 10.50 | 22.82 | 65.52 | 76.39 | 17.11 | 79.07 | 59.52 |
| Llama3-405B | 61.64 | 28.88 | 45.26 | 67.00 | 89.68 | 44.00 | 79.67 | 70.09 |
| LongCat-Flash | 58.31 | 33.33 | 45.82 | 64.38 | 78.55 | 39.56 | 82.11 | 66.15 |
| MiniMax-M2 | 57.18 | 32.29 | 44.74 | 66.00 | 65.37 | 37.11 | 82.33 | 62.70 |
| GLM-4.6 | 56.91 | 30.14 | 43.53 | 67.62 | 91.03 | 32.89 | 83.73 | 68.82 |
| Qwen3-next-80B | 45.32 | 20.67 | 33.00 | 62.69 | 84.98 | 32.89 | 78.89 | 64.86 |
| Qwen3-235B | 36.50 | 13.52 | 25.01 | 69.07 | 87.93 | 31.11 | 82.79 | 67.73 |
| GPT-OSS-120B | 41.43 | 12.21 | 26.82 | 68.63 | 70.48 | 21.67 | 76.95 | 59.43 |
| Kimi-K2 | 36.54 | 10.12 | 23.33 | 69.37 | 95.21 | 32.89 | 79.20 | 69.17 |

six state-of-the-art Models acting as orchestrators. We fit the analytical solution derived in Eq. (5) to the observed entropy evolution and cross-reference the resulting physical parameters with the actual step-wise execution accuracy (Figure 3). For a detailed discussion on the selection of the 6-step observation window and long-horizon results, refer to Appendix F.

We fit the proposed dynamics equation $\bar{H}(t)$ to the execution logs of each LLM. As illustrated in Figure 2, the model demonstrates a high degree of fidelity in capturing the distinct entropy evolution signatures of different agents, and the statistical significance of the regression was confirmed with $p < 0.005$ for all results. The decomposition curves in the figure clearly show how the total entropy is shaped by the competing forces of task oscillation (green dashed) and context accumulation (gray dotted). The specific fitted parameters are summarized in Table 4.

Our Mean-Field model successfully decouples the competing forces of task resolution and context loading. By treating system accuracy as an *order parameter* that is inversely correlated with thermodynamic entropy, we validate that the macroscopic parameters $\Theta = (A, \beta, \gamma, \omega)$ serve as predictive indicators of microscopic orchestrator performance.

*Table 4.* Parameter fitting results of mean-field dynamics equation.

| Model | $A_{\text{Task}}$ | $\gamma$ | $\omega$ | $\beta$ |
|---|---|---|---|---|
| Claude-4.5-Snt | $2.87 \pm 0.28$ | $0.94 \pm 0.12$ | $1.11 \pm 0.06$ | $0.87 \pm 0.08$ |
| Gemini-2.5-Pro | $3.10 \pm 0.08$ | $0.35 \pm 0.06$ | $0.55 \pm 0.06$ | $0.94 \pm 0.04$ |
| DeepSeek-V3.1 | $3.71 \pm 0.35$ | $2.13 \pm 0.11$ | $3.42 \pm 0.35$ | $1.37 \pm 0.04$ |
| Grok-4 | $4.47 \pm 0.05$ | $0.79 \pm 0.07$ | $0.91 \pm 0.02$ | $1.10 \pm 0.03$ |
| Qwen-3-Max | $7.65 \pm 0.12$ | $1.92 \pm 0.02$ | $1.73 \pm 0.01$ | $1.12 \pm 0.01$ |
| GPT-4.1 | $8.42 \pm 0.01$ | $1.52 \pm 0.01$ | $1.12 \pm 0.01$ | $0.94 \pm 0.00$ |

**Initial Complexity and The Phase Transition ($A_{\text{task}}, \omega$):** High values of task amplitude ($A_{\text{task}}$) and frequency ($\omega$) signify a system with high initial "temperature" in thermodynamic system and intense exploration. GPT-4.1 and Qwen-3-Max exhibit the highest initial entropy ($A_{\text{task}} = 8.415$ and $7.650$, respectively), while DeepSeek-V3.1 shows rapid hypothesis testing ($\omega = 3.418$).

This theoretical entropy surge correlates perfectly with a violent phase transition observed in empirical performance. Specifically, models identified with high active energy (Qwen-3-Max and DeepSeek-V3.1) suffer a precipitous collapse at Step 2 (e.g., DeepSeek drops from 92% to 22%). This confirms that the rapid expansion of the solution space, as modeled by our ODE, directly translates to a temporary loss of agent accuracy due to excessive uncertainty.

Entropy Dynamics Model Fitting Results

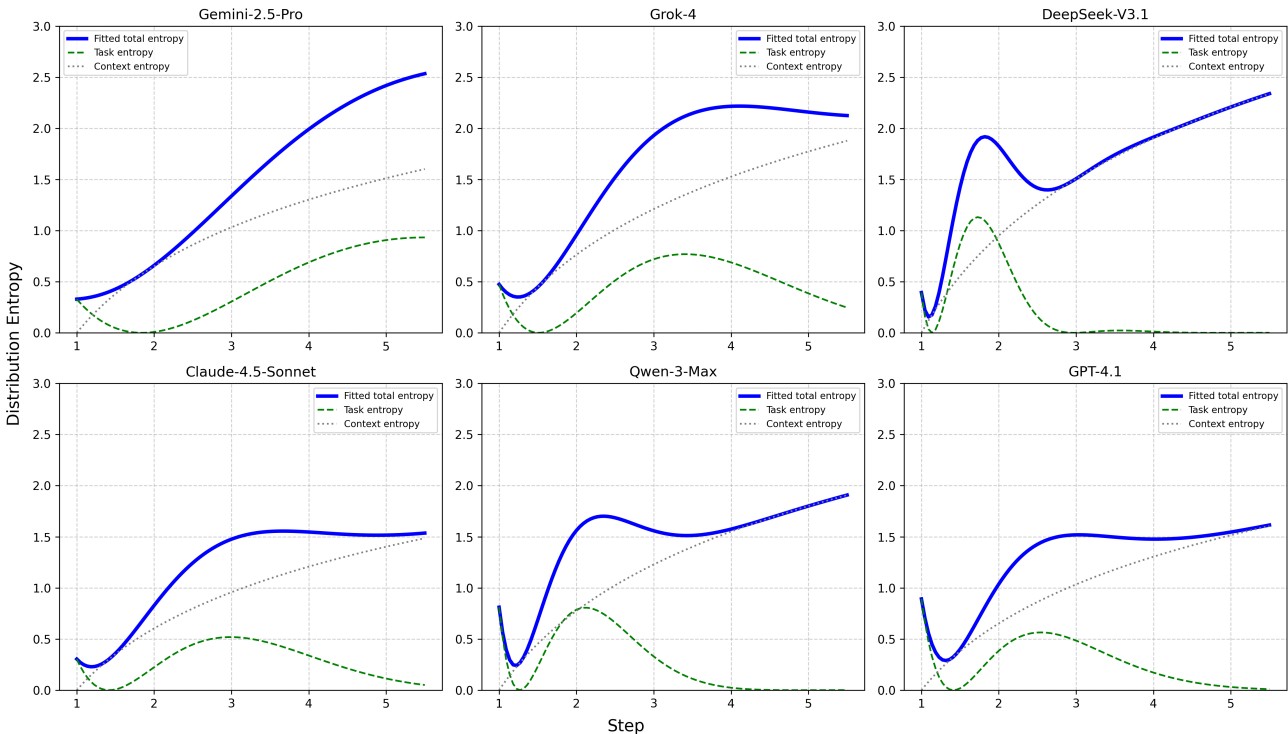

*Figure 2.* Experimental fitting results of the Mean-Field Entropy Dynamics model across six LLM orchestrators. The solid blue curve represents the total fitted entropy $\bar{H}(t)$, decomposed into the transient task-resolution component (green dashed) and the logarithmic context-loading component (gray dotted).

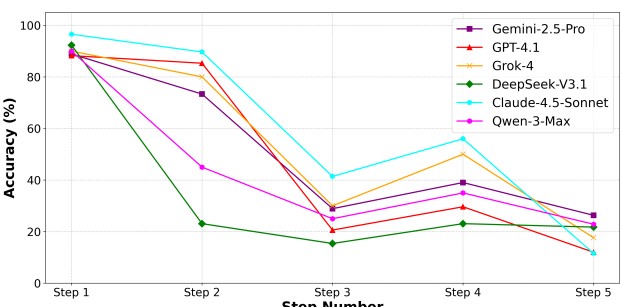

*Figure 3.* Step-Level accuracy of the LLM orchestrators.

**Contextual Sensitivity and Cognitive Endurance ($\beta$):**
The coefficient $\beta$ proxies the memory burden or sensitivity to history. DeepSeek-V3.1's high sensitivity ($\beta = 1.372$) contrasts sharply with Claude-4.5-Sonnet, which maintains the lowest baseline ($\beta = 0.871$).

Physically, Claude's low $\beta$ implies an efficient attention mechanism that filters irrelevant history. This is validated by its robust cognitive endurance in the accuracy analysis: unlike other models, Claude resists entropy accumulation, maintaining superior accuracy ($40 \sim 55\%$) even at Steps

3 and 4. Thus, $\beta$ effectively predicts an agent's ability to sustain order against a growing context stack.

**Damping Dynamics and Resolution Trajectory ($\gamma, \omega$):**
The decay rate $\gamma$ and oscillation $\omega$ determine how quickly uncertainty is resolved. DeepSeek-V3.1 exhibits high friction ($\gamma = 2.130$), collapsing superposition rapidly, whereas Gemini-2.5-Pro ($\gamma = 0.347, \omega = 0.551$) behaves as an overdamped system with prolonged exploration.

Crucially, the empirical data validate the oscillatory nature of our drift equation ($\cos \omega t$). A distinct accuracy recovery is observed at Step 4 for several models (notably Claude and Gemini). This corresponds to the *exploitation phase*—a temporary trough in task entropy where the system converges on a solution before the universal context load ($\beta \ln t$) dominates the trajectory.

### 4.4. Robustness of Entropy-Dynamics Parameters

We further examine whether the fitted entropy-dynamics parameters are specific to the main IWG benchmark by evaluating three complementary settings: controlled context, cross-domain transfer, and real-world transfer. All settings share the same fitting procedure and parameterization; the

IWG baseline refers to the raw environment used in the main benchmark. Table 5 summarizes the results.

**Controlled perturbation.** We compare the IWG baseline with two controlled variants. In the *Golden Context* setting, intermediate executor outputs are replaced with clean, ground-truth evidence, removing executor-side uncertainty and error propagation. Relative to the baseline ($A_{\text{task}} = 3.16$, $\gamma = 1.56$), Golden Context dramatically shrinks the task amplitude to 0.49 and the damping coefficient to 0.10, and it also lowers context sensitivity $\beta$. This confirms that a substantial fraction of the entropy fluctuation observed in the raw environment is driven by uncertainty accumulated through intermediate observations. In the opposite direction, *Exception Injection* (deliberately inserting noisy signals) pushes $A_{\text{task}}$ to 9.85 and raises $\gamma$ to 1.20, amplifying the very mechanisms that Golden Context suppresses. Crucially, the entropy trajectories across clean, noisy, and raw conditions preserve a stable overall shape and transition trend without collapsing into a qualitatively different regime. IWG thus captures the exception burden and noise sensitivity that characterize real-world multi-agent tool use, and our entropy-dynamics framework remains valid under such noisy workflows.

**Cross-domain transfer.** We next evaluate two structurally different domains. *File Operation* yields parameters ($A_{\text{task}} = 2.53$, $\gamma = 1.02$, $\beta = 0.32$) that remain close to the baseline, suggesting that file-centric tasks retain an orchestration pattern similar to that of the IWG benchmark. *MCP/API Calling* exhibits a distinct signature: exploration frequency $\omega$ rises to 2.93 while context sensitivity $\beta$ drops to near zero. This pattern aligns with the structured and concise nature of API interactions—once the correct call structure is identified, little long-context accumulation occurs, and the orchestrator can rapidly switch among candidate interfaces. The cross-domain variation thus reflects systematic differences in context length, rule structure, and tool feedback rather than idiosyncratic fitting.

**Real-world transfer.** Finally, we evaluate the fitted dynamics in a real-world setting without synthetic intermediate observations. Compared with the IWG baseline, real-world tasks show a substantially higher task amplitude ($A_{\text{task}} = 6.42$ vs. 3.16) and context sensitivity ($\beta = 0.68$ vs. 0.47), while exploration frequency decreases ($\omega = 1.03$ vs. 1.72). The increased $A_{\text{task}}$ likely originates from real-world noises—such as network latency and API instability—that trigger the agent's context management mechanisms, converting external disturbances into internal filtering and summarization efforts. Consequently, more cognitive budget is allocated to context consolidation and memory maintenance, which reduces the capacity for aggressive decomposition ($\gamma$) and exploration ($\omega$). Hence, the primary effect of realistic environments is not simply larger context but

*Table 5.* Fitted entropy-dynamics parameters across robustness settings. The IWG baseline corresponds to the raw IWG environment used in Section 4.3.

| Category | Setting | $A_{\text{task}}$ | $\gamma$ | $\omega$ | $\beta$ |
|---|---|---|---|---|---|
| Controlled Context | Golden Context | 0.49 | 0.10 | 1.45 | 0.16 |
| | Exception Injection | 9.85 | 1.20 | 1.23 | 0.32 |
| Cross-Domain | File Operation | 2.53 | 1.02 | 1.63 | 0.32 |
| | MCP/API Calling | 0.49 | 0.46 | 2.93 | 0.01 |
| Real-world Transfer | Real-world | 6.42 | 1.14 | 1.03 | 0.68 |
| | Baseline | 3.16 | 1.56 | 1.72 | 0.47 |

harder information management and execution under noise. Notably, the fitted values for the real-world setting and the IWG baseline remain on the same order of magnitude, indicating that IWG provides a reasonable approximation of real-world orchestration difficulty.

Overall, these results indicate that the proposed entropy-dynamics parameters remain interpretable across controlled perturbation, domain shift, and real-world transfer. Rather than being benchmark-specific curve-fitting coefficients, the parameters serve as compact behavioral descriptors for diagnosing how the orchestration difficulty systematically varies across evaluation regimes.

### 4.5. The "Reasoning Trap" in Heavy Thinking Models

A counterintuitive finding from our benchmark (Table 3) is the suboptimal performance of reasoning-centric models. Despite their superior logical capabilities in closed-form evaluations, they fail significantly as Orchestrators. We term this phenomenon the **Reasoning Trap.**

Unlike standard instruction-following models that act as Routers, reasoning models generate extensive internal Chain-of-Thought (CoT) sequences before emitting actions. In an orchestration step, the prompt includes the user query, system instructions, and verbose logs from multiple executors. Reasoning models often generate substantial internal tokens in the reasoning content. The massive self-generated CoT effectively squeezes the attention budget available for external signals (Liu et al., 2024). Furthermore, excessive reasoning steps do not necessarily lead to convergence but can introduce noise that distracts the model from the original instruction constraints (Wei et al., 2025b).

To quantify this, we compared GPT-5 and Qwen3-Max using their respective reasoning effect settings as GPT-5-Low/High (OpenAI, 2025) and Qwen3-Max/Qwen3-Max-NoThink) (Yang et al., 2025a). In addition to the evaluation metrics mentioned above, we introduce a token usage metric: Token Efficiency, which represents the proportion of the maximum context budget that remains unconsumed after inference, calculated as $1 - (N_{consumed}/N_{max})$.

As illustrated in Figure 4, suppressing Overthinking leads to

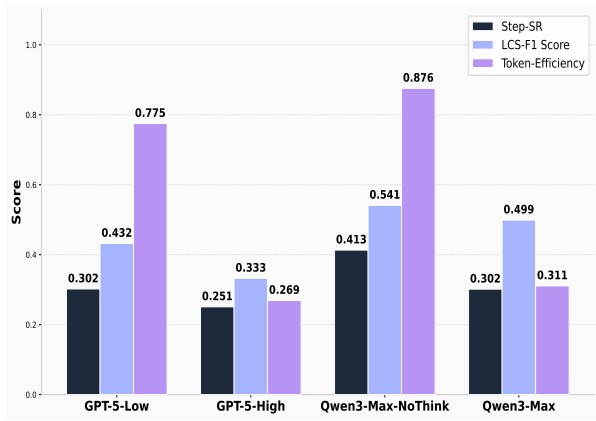

*Figure 4.* **Impact of Reasoning Depth on Orchestration Performance.** Intra-model comparisons reveal that Light Thinking variants (GPT-5-Low, Qwen3-Max-NoThink) significantly outperform their reasoning-heavy counterparts.

substantial performance gains within each model family. For the **GPT-5** series, shifting from the verbose GPT-5-High to the constrained GPT-5-Low reduces token usage by approximately 69% while improving the Step Success Rate from $0.251$ to $0.302$. Similarly, the standard Qwen3-Max exhibits typical overthinking symptoms, while the **NoThink** variant, which bypasses internal reasoning, achieves a dramatic efficiency gain and boosts the Step-SR to $0.413$. These empirical results validate the **Context Squeezing** hypothesis. In reasoning-centric modes, the model's attention budget is saturated by self-generated tokens, diluting the influence of external instructions. While the internal tokens increase, the model becomes absorbed in its own reasoning trace, losing focus on the rigid constraints provided in the system prompt. Adopting Light Thinking strategies prevents the model from hallucinating unnecessary complexities, thereby enhancing the orchestration success.

## 5. Related Work

### 5.1. Multi-Agent Collaboration Frameworks

Research in MAS has bifurcated into two primary coordination patterns. **Decentralized/Cooperative Systems:** Early works like CAMEL (Li et al., 2023) introduced role-playing frameworks where agents communicate peer-to-peer (P2P) without a central supervisor. Similarly, Park et al. demonstrated emergent social behaviors in agents with decentralized memory streams. While robust, these systems can suffer from infinite dialogue loops and lack of goal-directed convergence. Gao et al. directly compare single-agent and multi-agent architectures, finding that MAS advantages over SAS can diminish as frontier models improve, and propose a hybrid cascading design to balance capability and cost.

### 5.2. Centralized/Hierarchical Systems

To ensure goal alignment, frameworks like MetaGPT (Hong et al., 2023) and AutoGen (Wu et al., 2023) formalized Standard Operating Procedures (SOPs) encoded into a manager agent. In these systems, an Orchestrator maintains the global state and dispatches tasks. Magentic-One (Research, 2024) advances this by integrating a dynamic task ledger for the Orchestrator. Our work focuses specifically on this hierarchical topology. Webpilot (Zhang et al., 2025b) further refines this by employing an planner that integrates a recursive planning-and-acting loop, enabling strategic exploration and dynamic plan adjustment in open-ended web environments. Similarly, AgentOrchestra (Zhang et al., 2025a) proposes a vertical hierarchy where the Orchestrator functions as a conductor, transforming high-level goals into executable sub-tasks through structured decomposition.

### 5.3. Planning and Reasoning in LLMs

The core competence of an Orchestrator is planning. Approaches like Tree of Thoughts (ToT) (Yao et al., 2023a) attempt to structure the reasoning process as a search problem. ReAct (Yao et al., 2023b) introduces dynamic interaction and goal decomposition to bridge reasoning with action. Furthermore, Reflexion (Shinn et al., 2023) incorporates iterative self-correction through linguistic feedback. Tang et al. (Tang et al., 2025) argue that task complexity, characterized by reasoning depth and capability width, is central to evaluating LLM-based MAS, rather than relying on surface-level task categories. Recent evaluations suggest that while LLMs are becoming better at using function calling, their ability to maintain long-term logical consistency in a dynamic environment remains fragile (Valmeekam et al., 2023). Yang et al. (Yang et al., 2025b) further study multi-agent debate as a form of test-time scaling, showing that its effectiveness is conditional on task difficulty, model capability, and agent diversity. Our experiment empirically validates this fragility in the context of multi-agent orchestration.

## 6. Conclusion

In this work, we shifted the analytical focus of Multi-Agent Systems from the capabilities of individual agents to the dynamics of the Orchestrator. Through a novel Mean-Field Entropy Dynamics perspective, we successfully modeled the orchestration process as a superposition of oscillatory task exploration and logarithmic context accumulation. Our theoretical model, validated against empirical data from SOTA LLMs, provides a physical interpretation of MAS. By providing the physical mechanisms underlying the Orchestrator and quantifying systemic uncertainty, these findings offer insights for the architectural design development of Multi-Agent Systems in prospective research.

# Acknowledgements

We thank the anonymous reviewers for their insightful feedback and suggestions. This work is supported by the National Natural Science Foundation of China (No. 62576163, 62376120) and the Fundamental Research Funds for the Central Universities (No. 2026300382).

# Impact Statement

Our work advances the theoretical understanding and architectural design of Multi-Agent Systems by establishing a Mean-Field Entropy Dynamics framework to diagnose orchestration fragility. While transitioning to autonomous orchestration enhances problem-solving, we acknowledge the inherent risks of system collapse and unreliability, particularly the "Reasoning Trap" where heavy-thinking models lose focus on strict constraints. To mitigate these risks and ensure rigorous evaluation, we employed the Inverse Workflow Generation pipeline, which utilizes a multi-tier Validation Committee with human-in-the-loop verification to guarantee the logical validity and safety of generated agent trajectories. Our findings advocate for efficient, "Instant Breadth-First Thinking" orchestration strategies, emphasizing that future Multi-Agent System deployment must prioritize predictable, physically interpretable stability metrics over raw reasoning power to prevent uncontrolled context dispersion and ensure safe system behavior.

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

# A. MAS Failure Attribution Analysis

We implement and evaluate for **GPT-4o** (Hurst et al., 2024) as Orchestrator four representative multi-agent systems operating under the Orchestrator-Executor topology. The configurations are detailed in Table 6.

The systems were tested on a diverse suite of benchmarks including GAIA (Mialon et al., 2023), BrowserComp (Wei et al., 2025a), AssistantBench (Yoran et al., 2024), OCR-VQA (Mishra et al., 2019), HotpotQA(Yang et al., 2018), HumanEval (Chen, 2021) and DeepResearchBench (Du et al., 2025). We employed **GPT-4** (Achiam et al., 2023) and **GPT-4o** (Hurst et al., 2024) as the backbone model for executor agents to ensure their sufficient capabilities.

*Table 6.* System Configurations for Failure Attribution Study

| System | Reference | Our Implement (Orchestrator + Executors) |
|---|---|---|
| **Deep Research** | STORM (Shao et al., 2024) | OnlineRetriever, Writer, Reporter |
| **Agent Coder** | Magentic-One (Research, 2024) | WebSurfer, FileSurfer, Coder, Terminal |
| **Web Browser** | Plan&Actor (Erdogan et al.) | VisionAgent, WebGUIOperator |
| **Agentic RAG** | Hybrid RAG (Papageorgiou et al., 2025) | NewsSearcher, KGRetriever, Summarizer |

The results of the failure attribution analysis are illustrated in Figure 5. Failures were manually annotated and classified into *Orchestrator Failure* (planning/routing errors) or *Executor Failure* (tool use errors).

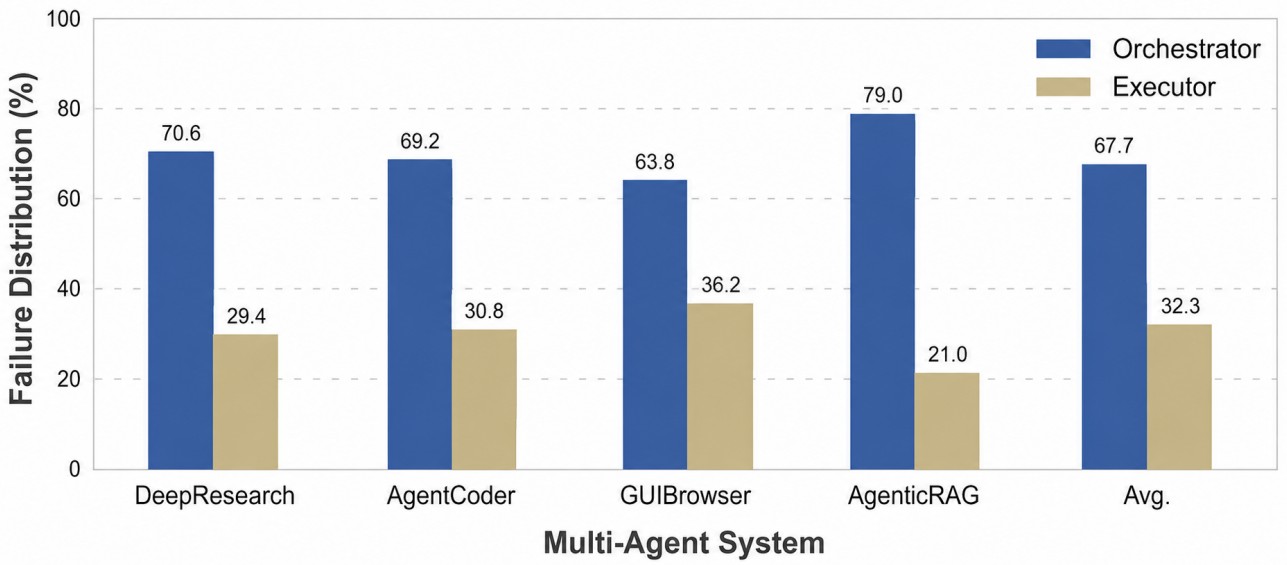

*Figure 5.* Pre-Experiment Results: Task Failures Analysis. The chart contrasts the failure contribution of the Orchestrator (Blue) vs. the Executor (Gold) across four systems.

The data reveals a consistent pattern of **Orchestrator dominance in failure rates** across all four tested systems:

The global trend of pre-experiment results point the finger at Orchestrator. The Orchestrator was responsible for a mean of **67.7%** of all failures, whereas Executors accounted for only 32.3%. This indicates that the critical weakness lies in the management layer, not the execution layer. The Agentic RAG system exhibited the highest Orchestrator failure rate (79.0%) in the high-complexity retrieval tasks. In scenarios requiring multi-hop reasoning over retrieved knowledge graphs, the Orchestrator frequently failed to synthesize conflicting reports from the Retrievers and make unsatisfactory retrieval strategies. On the other side, the Vision Web Browser system had the highest Executor failure rate (36.2%), attributed to the invalid image urls and unexpected browser content. However, even here, the Orchestrator caused the majority of failures.

Our case analysis of error traces suggest that, in certain instances, the Orchestrator demonstrates the capacity to detect anomalies in the Executor's response and consequently reconfigures the task sequence. However, the Orchestrator lost track of the task state in common station, leading to repetitive loops or premature termination.

Within tolerable limits, the Orchestrator can robustly absorb the Executor's failures. Crucially, however, this error correction mechanism converts execution glitches into additional reasoning pressure on the central node, which partly explains why empirical traces show a higher apparent failure rate for the Orchestrator than for the Executor.

Based on these empirical findings, we hypothesize a distinction in failure mechanics: Executor failures typically arise when task demands exceed the system's absolute ability boundary, whereas Orchestrator failures appear to occur when the environment information entropy surpasses the Orchestrator's epistemic threshold.

## B. Theoretical Derivation of Entropy Dynamics

In the main text (Section 2), we modeled the evolution of the macroscopic scheduling entropy $\bar{H}(t)$ as the superposition of a focusing operator $\mathcal{F}_{\text{task}}$ and a dispersion operator $\mathcal{D}_{\text{context}}$. Here, we provide the rigorous mathematical derivation for the functional forms of these operators, grounding our phenomenological model in optimization theory and statistical mechanics.

### B.1. Derivation of the Focusing Operator ($\mathcal{F}_{\text{task}}$)

We posit that the Orchestrator's search for the optimal executor subset is analogous to an optimization process on a probability manifold. Let $\theta_t \in \mathbb{R}^n$ represent the continuous state of the policy network at time $t$. The objective is to minimize a "cognitive potential" function $V(\theta)$, where the minimum $\theta^*$ corresponds to the optimal agent allocation (lowest entropy).

**Step 1: The Inertial Assumption.** Unlike memoryless Gradient Descent (GD), where $\theta_{k+1} = \theta_k - \eta \nabla V(\theta_k)$, Large Language Models (LLMs) maintain a history of past tokens. The attention mechanism effectively creates a "memory" of previous states, resisting abrupt policy changes. This introduces an inertial term, equivalent to **Momentum** in optimization. The discrete update rule follows Nesterov's Accelerated Gradient (NAG):

$$\theta_{k+1} = y_k - \eta \nabla V(y_k), \quad y_k = \theta_k + \frac{k-1}{k+2}(\theta_k - \theta_{k-1}) \tag{7}$$

**Step 2: Continuous Limit as a Damped Oscillator.** Following the seminal work by Su, Boyd, and Candès (Su et al., 2016), we analyze the behavior of Eq. 7 as the step size $\sqrt{\eta} \to 0$. The discrete difference equation converges to a second-order Ordinary Differential Equation (ODE):

$$\ddot{\theta}(t) + \frac{3}{t}\dot{\theta}(t) + \nabla V(\theta(t)) = 0 \tag{8}$$

This equation describes a particle with unit mass moving in a potential $V(\theta)$ subject to time-dependent friction.

For the specific case of an Orchestrator with a finite context window, the friction does not decay to zero but stabilizes at a constant "learning rate" determined by the model's effective attention span. We thus approximate the dynamics with the classic **Heavy Ball** friction model (constant damping $\gamma$):

$$m\ddot{\theta}(t) + \gamma\dot{\theta}(t) + k(\theta(t) - \theta^*) = 0 \tag{9}$$

where $k$ is the curvature of the potential (task difficulty).

**Step 3: Solution and Entropy Envelope.** For a challenging task (high difficulty $k$), the system is underdamped ($\gamma^2 < 4mk$), leading to the solution:

$$\theta(t) - \theta^* \propto e^{-\frac{\gamma}{2m}t}\cos(\omega t + \phi) \tag{10}$$

The macroscopic entropy $\bar{H}(t)$ measures the spread or uncertainty of the distribution. In a physical system, entropy correlates with the envelope of the system's kinetic and potential energy (the deviation from equilibrium). Therefore, the entropy reduction component follows the amplitude envelope of the oscillation:

$$\mathcal{F}_{\text{task}}(t) = \frac{d}{dt}\left(A_{\text{task}}e^{-\frac{\gamma}{2m}t}\sin(\omega t + \phi)\right) \tag{11}$$

This rigorously justifies the use of the damped sine wave to model the Orchestrator's "hunting behavior" during reasoning.

## B.2. Derivation of the Dispersion Operator ($\mathcal{D}_{\text{context}}$)

The dispersion operator models the entropy growth caused by the accumulation of context $\mathcal{C}_t$. We treat the Orchestrator's probability field as a wave packet evolving in the presence of noise.

**Step 1: Context as a Diffusive Process.** According to (Wibisono, 2018), sampling dynamics can be viewed as gradient flows in the space of probability measures. The accumulation of context introduces high-dimensional noise, acting as a "heat bath." The evolution of the probability density $\rho(x, t)$ is governed by the Fokker-Planck equation with a diffusion coefficient $D$:

$$\frac{\partial \rho}{\partial t} = \cdots + D\nabla^2 \rho \tag{12}$$

Focusing solely on the diffusion term, the variance $\sigma^2(t)$ of the wave packet grows linearly with time (Einstein's relation):

$$\sigma^2(t) = \sigma_0^2 + 2D \cdot t \approx 2Dt \quad \text{(for large } t) \tag{13}$$

**Step 2: Entropy-Variance Relationship.** For a canonical distribution (e.g., Gaussian), the differential entropy $H(t)$ is directly related to the variance. As established in standard information theory (Shannon, 1948):

$$H(t) = \frac{1}{2}\ln(2\pi e\sigma^2(t)) \tag{14}$$

**Step 3: Logarithmic Scaling.** Substituting the linear variance growth (Eq. 13) into the entropy definition:

$$\begin{aligned} H(t) &= \frac{1}{2}\ln(2\pi e \cdot 2Dt) \\ &= \frac{1}{2}\ln(t) + \underbrace{\frac{1}{2}\ln(4\pi eD)}_{\text{Constant}} \end{aligned}$$

Defining the dispersion rate $\beta = 1/2$ (or scaling $D$ accordingly), we arrive at the logarithmic scaling law:

$$\mathcal{D}_{\text{context}}(t) \to \frac{d}{dt}\left(\beta \ln(t+1)\right) = \frac{\beta}{t+1} \tag{15}$$

**Alternative Perspective: Attention Degeneration.** This result is consistent with the behavior of Transformer self-attention. As shown by Dong et al. (Dong et al., 2021), without strong constraints, deep attention layers tend to degenerate toward a uniform distribution (maximum entropy) as the sequence length $L$ increases. Since $L \propto t$, and the entropy of a uniform distribution is $\ln L$, we again obtain $H \sim \ln t$. The diffusion model is thus a robust generalization valid for both physical particles and attention mechanisms.

### B.3. Synthesis of the Macroscopic Equation

Combining the solutions from Appendix B.1 and B.2, we obtain the final governing equation used in the methodology:

$$\bar{H}(t) = \underbrace{A_{\text{task}}e^{-\gamma t}\sin(\omega t + \phi)}_{\text{from Momentum Optimization}} + \underbrace{\beta \ln(t+1)}_{\text{from Information Diffusion}} + H_0 \tag{16}$$

This derivation confirms that our model parameters $(\gamma, \omega, \beta)$ are not arbitrary regression coefficients, but physical quantities corresponding to the *friction*, *natural frequency*, and *diffusion rate* of the Orchestrator's reasoning process.

## C. Pipeline Implementation and Case Study of IWG

To bridge the data gap, we propose the IWG (Inverse Workflow Generation) pipeline for synthesizing complex Agent Task data with clear step-level validation checks. As illustrated in Figure 1, we utilize a running example from the seed data *What year was the director of The White Ribbon born?* to a agent task about a playlist operation.

The IWG operates on the principle of **Duality** and **Inverse Synthesis**. Unlike conventional forward-chaining Agent execution—which proceeds from initial environment to final result, IWG reverses the process. This system utilizes a

multi-agent architecture to process high-quality, pre-verified answers (Seed Data) and generates the complex, multi-step execution trajectory and the necessary environmental informations required to reach those answers. This inverse approach ensures that the resulting benchmark data is inherently task-solvable, verifiable, and contains the explicit intermediate steps necessary for entropy measurement. The final synthesized task instance follows a prompt chaining workflow pipeline defined by a Directed Acyclic Graph (DAG) pattern.

The IWG functions as a multi-agent pipeline comprising three primary components, each fulfilling a distinct role in the inverse synthesis chain.

### C.1. The Scout Agent (Inverse Planning and Decomposition)

The Scout Agent initiates the workflow. Crucially, it takes two inputs: the **Seed Data** and the **Target Multi-Agent System Configuration**. Its role is not only to decompose the task in reverse, but to ensure the decomposition fully leverages and benchmarks the specific capabilities of the available agents.

**Capability-Aware Inverse Analysis:**  The Scout Agent first profiles the member agents (e.g., Vision Agent, Entity Retriever, GUI Operator) to understand their specific tools and triggers. It then analyzes the Seed Data to find logical entry points for these agents and determines the sequence of executor actions.

**Task Decomposition:**  It understands and breaks down the overall task content, generating **Task Marks** and determining the capabilities needed by the Executor Agent in the final benchmark. The Scout breaks the workflow into task marks $(M_0, M_1)$ that map directly to agent capabilities. It assigns the visual task for poster identification to the Vision Agent and the information retrieval task for director documentation to the Entity Retriever.

**Task Extend:**  Although the original Seed Data does not explicitly involve a playlist operation, the Scout detects the presence of a **GUI Operator** in the multi-agent system. To benchmark this agent, the Scout proactively generates a logical follow-up task $(M_2)$: *Add the movie to the playlist*. This ensures that the **[app_operation]** capability is exercised and validated within the workflow, even if it wasn't part of the initial static QA pair.

### C.2. The Wrapper Agent (Environment Synthesis and Task Encapsulation)

The Wrapper Agent acts as the environmental generator, converting the Scout's plan into a concrete interaction environment with verifiable outputs.

**Environment Synthesis ($EI$):**  The Wrapper generates the specific inputs and simulated tool outputs for each turn based on those Scout Marks. For instance, in **Query_Turn2**, the Wrapper synthesizes the Environmental Information ($EI_1$) simulating an Entity Retriever's output: *A film by Michael Haneke set in rural Germany....* This provides the context the agent needs to proceed.

**Checkpoint Generation:**  Crucially, the Wrapper extracts key information to form **checkpoints** for automated evaluation. Instead of relying on ambiguous LLM scoring, we verify specific strings like **[checkpoint_0]** *The White Ribbon*. For tasks that cannot be matched with string labels, a 1-shot customized verification method is used, such as **[checkpoint_3]** using API(GET https://filmchain/...) to query data and verify GUI operation results.

### C.3. The Validation Committee Agent Group (Quality Control)

The Validation Committee is an indispensable Agent Group responsible for enforcing data quality and logical consistency within the generated benchmarks. Specifically, we implement a **Three-Tier Validation Protocol**. The core validation mechanism involves verifying whether the checkpoints can be correctly deduced solely from the Environmental Information ($EI$) generated by the Wrapper.

**Tier 1 (Solvability Check):**  An open-source model (Qwen2.5-32B) acts as the first examiner. It attempts to execute the task steps using the synthesized $EI$. The data instance is retained only if the model can successfully infer the values of the pre-defined checkpoints, proving that the generated environment provides sufficient context.

**Tier 2 (Consistency Check):**   Instances passing Tier 1 are re-evaluated by a closed-source model (gpt4o-mini). This step confirms that the reasoning path is robust and not an artifact of a specific model architecture. We require that both models successfully derive the checkpoints without ambiguity.

**Tier 3 (Human-in-the-Loop):**   Only data that has achieved consensus across both model tiers enters the final stage. Human experts review the factual correctness and validate the overall task logic, ensuring the validity of the final benchmark.

## D. Metric Definitions and Calculation Protocols

To rigorously quantify the performance of the LLM Orchestrator, we define six metrics based on the specific topological properties of the IWG data. We use *The White Ribbon* case (Figure 2) to illustrate these calculations.

### D.1. System-Level Metrics

#### D.1.1. 1. LCS-F1 (AGENT SEQUENCE STRUCTURAL SIMILARITY)

This metric evaluates the orchestrator's planning logic by comparing the predicted sequence of agent calls against the ground truth workflow synthesized by IWG.

Let the ground truth agent sequence be $S_{gold} = [a_1, a_2, \ldots, a_n]$ and the predicted sequence be $S_{pred} = [\hat{a}_1, \hat{a}_2, \ldots, \hat{a}_m]$. We compute the Longest Common Subsequence ($LCS$) between these two lists.

$$Precision = \frac{|LCS(S_{gold}, S_{pred})|}{|S_{pred}|}, \quad Recall = \frac{|LCS(S_{gold}, S_{pred})|}{|S_{gold}|} \tag{17}$$

$$LCS\text{-}F1 = \frac{2 \cdot Precision \cdot Recall}{Precision + Recall} \tag{18}$$

**Example:** In Figure 2, the gold agent sequence is `[Vision, EntityRetriever, EntityRetriever, GUIOp]`. If the model skips the date check and outputs `[Vision, EntityRetriever, GUIOp]`, the LCS length is 3. $Recall = 3/4 = 0.75$.

#### D.1.2. 2. TASK SUCCESS (TS)

Task Success is a stringent binary metric. A task is considered successful if and only if **all** pre-defined checkpoints in the trajectory are correctly matched.

$$TS = \prod_{i=1}^{K} \mathbb{I}(\text{match}(\hat{c}_i, c_i^{gold})) \tag{19}$$

where $K$ is the total number of checkpoints in the task, and $\mathbb{I}$ is the indicator function. **Example:** The task in Figure 2 has 4 checkpoints. If the agent correctly identifies *The White Ribbon* ($c_0$) and *Michael Haneke* ($c_1$) but fails to retrieve the birth year **1942** ($c_2$), then $TS = 0$, even if the final playlist operation ($c_3$) is correct.

### D.2. Orchestrator-Level Metrics

#### D.2.1. 3. STEP SUCCESS RATE (STEP-SR)

Step-SR offers fine-grained granularity by calculating the micro-average accuracy of individual checkpoint matches across the entire dataset.

$$Step\text{-}SR = \frac{\sum_{\mathcal{T}} \sum_{i \in \mathcal{T}} \mathbb{I}(\text{match}(\hat{c}_i, c_i^{gold}))}{\sum_{\mathcal{T}} |C_{\mathcal{T}}|} \tag{20}$$

This allows us to credit models that can partially solve complex tasks (e.g., getting 3 out of 4 steps correct).

#### D.2.2. 4. EXCEPTION HANDLING F1 (EH-F1)

We evaluate robustness by injecting synthetic exceptions (e.g., *Network Timeout 404*) at random steps. The IWG dataset contains a gold recovery plan ($R_{gold}$) for each exception. We compare the model's generated recovery action ($R_{pred}$) against $R_{gold}$.

$$EH\text{-}F1 = F1\_Score(R_{pred}, R_{gold}) \tag{21}$$

The score is calculated based on the semantic overlap or classification match of the recovery strategy (e.g., Retry, Switch Others, Abort).

### D.2.3. 5. FAITHFULNESS (CONTEXT UTILIZATION RECALL)

Faithfulness measures whether the orchestrator correctly utilizes information gained from the previous step. It is defined as the **Recall** of the key entities from the previous turn's checkpoint ($c_{t-1}$) within the current turn's generated query ($q_t$).

$$Faithfulness_t = \frac{|tokens(c_{t-1}) \cap tokens(q_t)|}{|tokens(c_{t-1})|} \tag{22}$$

**Example:** In Query_Turn2 (Figure 2), the model asks *Who directed it?*. The **it** must refer to the movie identified in Turn 1. If $c_0$ = *The White Ribbon* and the generated query explicitly mentions *The White Ribbon* or implicitly references it correctly (resolved via coreference resolution), the recall is high. If the model hallucinates a different movie, recall drops.

### D.2.4. 6. CONSISTENCY (GLOBAL ALIGNMENT)

Consistency measures the overall semantic similarity between the model's entire execution log and the IWG-synthesized gold trajectory.

$$Consistency = \cos(\mathbf{E}(Trajectory_{pred}), \mathbf{E}(Trajectory_{gold})) \tag{23}$$

where $\mathbf{E}$ denotes the vector representation from the `text-embedding-v4` model. This captures whether the orchestrator's reasoning path globally aligns with the intended logic, even if exact wordings differ.

## E. System Entropy Distribution Analysis

To investigate the stability of the orchestration process over time, we aggregated the executor agent selection distribution across all tested models for the first six planning steps. Figure 6 visualizes the frequency of calls to specific executor agents ($e_1$ to $e_7$) at each step. The statistical trend and detail analysis has been reported in Figure 6.

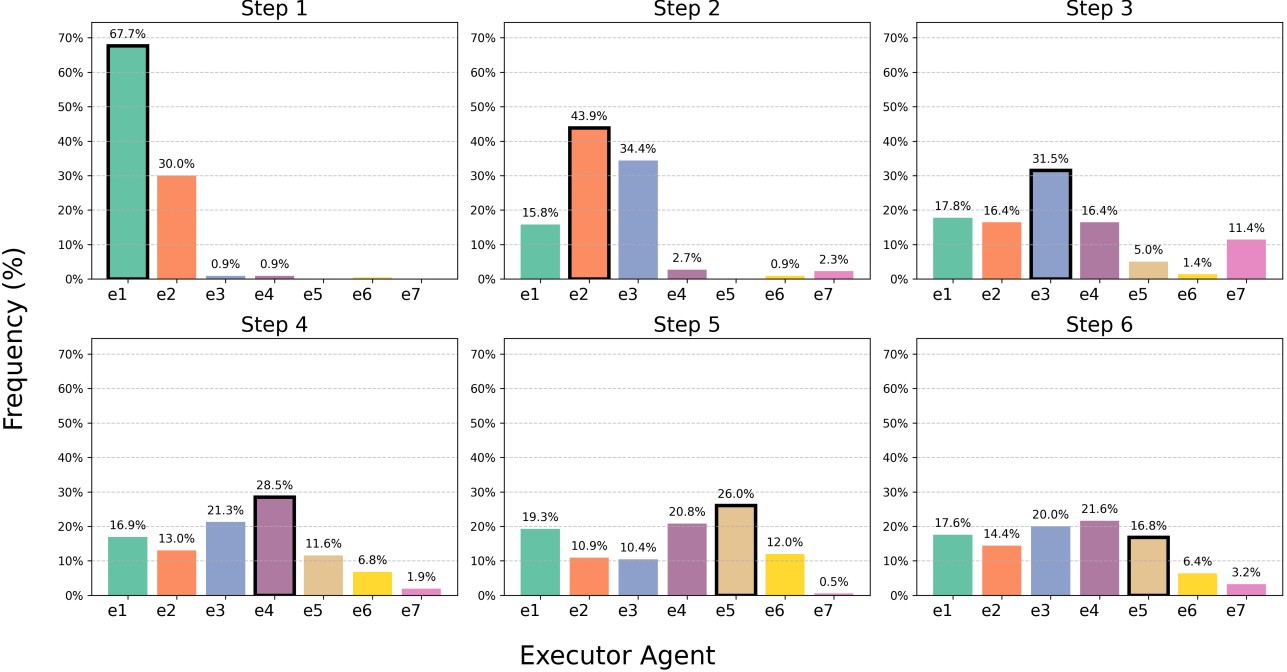

*Figure 6.* Experimental fitting results of the Mean-Field Entropy Dynamics model across six LLM orchestrators. The red dots indicate actual empirical data. The blue curve represents the total fitted entropy $\bar{H}(t)$, decomposed into the transient task-resolution component (green dashed) and the logarithmic context-loading component (gray dotted).

The statistical trend reveals a critical phenomenon of **System Entropy Gain** and a corresponding **Decay in Determinism**.

### E.1. The Entropy Increase Phenomenon

As illustrated in Step 1, the system exhibits high determinism (Low Entropy), with a dominant preference for agent $e_1$ (67.7%) and $e_2$ (30.0%), indicating a highly standardized entry point for the tasks. However, as the planning trajectory extends, the distribution rapidly flattens:

- **Transition Phase (Steps 2-3):** The focus shifts but begins to disperse. By Step 3, while $e_3$ leads with 31.5%, the usage of other agents ($e_1, e_2, e_4$) remains significant, suggesting a divergence in solution paths.

- **High Entropy Phase (Steps 4-6):** The distribution approaches uniformity. In Step 6, the variance between agents is minimal (e.g., $e_4$ at 21.6% vs. $e_3$ at 20.0% vs. $e_1$ at 17.6%).

### E.2. Implications of Determinism Reduction

This trend quantifies the challenge of long-horizon orchestration. The entropy increase phenomenon implies that as the conversation history accumulates, the decision space expands, and the consensus among models fractures.

Mathematically, this represents a cumulative accumulation of uncertainty. In the later steps (Step 4+), the *Context Pressure* discussed in previous sections exacerbates this uncertainty. The lack of a dominant executor in later steps suggests that even minor hallucinations or deviations in early steps propagate downstream, forcing models into diverse, and often suboptimal, recovery paths. This effectively reduces the theoretical upper bound of Task Success rates for long-chain tasks.

## F. Observation Window and Effective Cognitive Horizon

While the theoretical maximum step limit for our experiments was set to $k_{max} = 20$, our primary empirical analysis and parameter fitting focus on the first 6 orchestration steps ($t \in [1, 6]$). This selection is not arbitrary but is grounded in the task density of our benchmark and the entropy saturation phenomenon observed in current LLMs.

### F.1. Orchestration Step Density

It is crucial to distinguish an *Orchestrator Step* in our topology from a standard *Chain-of-Thought Step*.

- **Atomic vs. Macro Actions:** In standard CoT, a step might be a single arithmetic operation. In our Multi-Agent System, a single Orchestrator step ($t \rightarrow t + 1$) involves the formulation of a high-level directive, the delegation to a specialist Executor (e.g., DeepResearch Agent reading papers), and the digestion of the Executor's verbose output.

- **Complexity Equivalence:** Consequently, a 6-step trajectory in the IWG benchmark represents a high-density workflow equivalent to 20-30 atomic reasoning steps in single-agent benchmarks.

### F.2. Entropy Saturation and the Chaotic Regime

Our Entropy Dynamics model reveals a critical system boundary. As visualized in Figure 6 of the main text, the distribution of agent selection probabilities $p_k$ undergoes a transition:

1. **Ordered Regime ($t < 4$):** The system exhibits distinct preferences for specific tools, indicating clear planning logic.

2. **Transition Phase ($t = 4 \sim 5$):** The Task Entropy component oscillates, and Context Load begins to dominate.

3. **Chaotic Regime ($t \geq 6$):** The system reaches a state of *Entropy Saturation*. The agent selection distribution approaches uniformity ($p_i \approx 1/n$), and the Task Success rate for complex queries drops below 20% for all tested models.

In addition, the occurrence time of these three stages presents different intervals in different configurations according to task complexity and system design.

### F.3. Long-Horizon Generalization

To verify that the core entropy-success relationship persists beyond the primary 6-step setting, we conducted a long-horizon generalization test using 12-step tracking under a memory-compression condition. The mean-field dynamics equation was

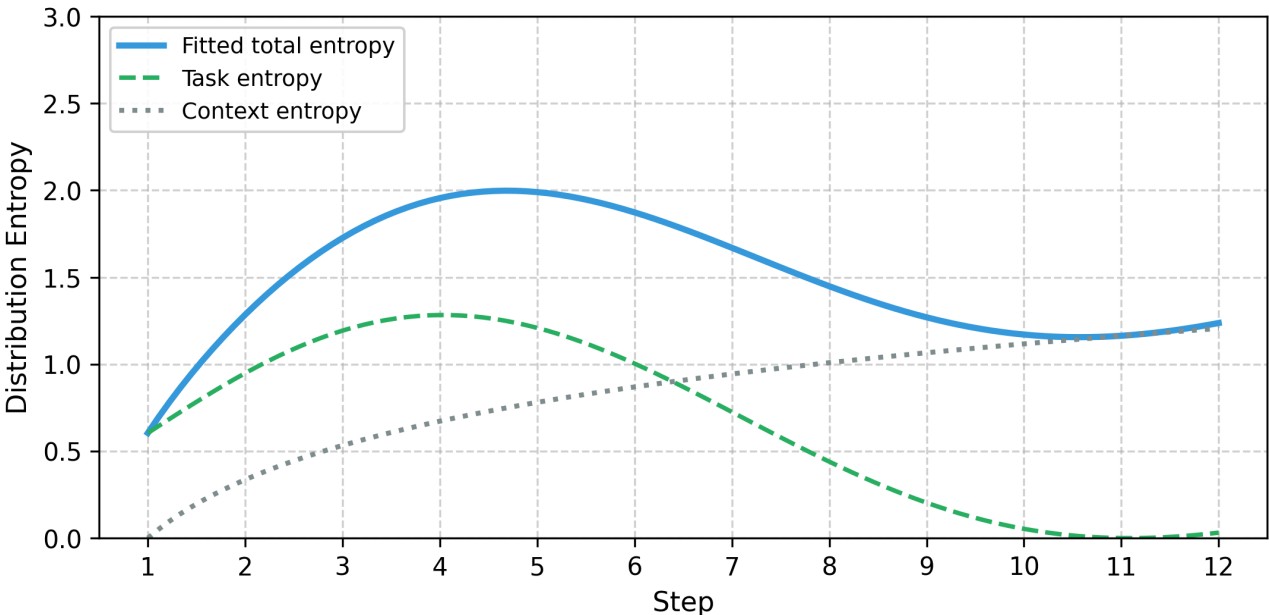

*Figure 7.* Experimental fitting results of 12-step long task setting. The solid blue curve represents the total fitted entropy $\bar{H}(t)$, decomposed into the transient task-resolution component (green dashed) and the logarithmic context-loading component (gray dotted).

refitted to the extended trajectories, yielding parameters $A_{\text{Task}} = 0.89$, $\gamma = 0.09$, $\omega = 0.50$, $\beta = 0.49$. These values remain fully consistent with the theoretical framework: the reduced task amplitude and damping reflect the system's transition toward a saturated, low-energy regime, while the contextual sensitivity $\beta$ stays within the expected range.

Critically, even in this extended regime, entropy maintains a significant negative correlation with step-level success (Pearson $r = -0.73$, $p < 0.05$). As shown in Figure 7 (the corresponding long-horizon result you have obtained), the predicted entropy continues to track the decay of task accuracy, confirming that the inverse relationship between thermodynamic entropy and orchestrator performance is not an artifact of the 6-step window but a robust dynamical signature. This finding supports the validity of our mean-field model as a general descriptor of multi-agent orchestration, capable of bridging short-term phase transitions and long-term cognitive degradation under memory constraints.

## G. System Setting

### G.1. Complete Case of the Orchestration MAS Workflow

The following table contrasts two orchestration paradigms applied to the *Miss Chris Audio Request* scenario.

| Step | A complete case of an Orchestrator-MAS handling daily issues for financial professionals |
|---|---|
| 1 | **Orchestrator-Plan:**
**thought_process:** Initial state. Parse audio to get intent.
**task_board_updates:** [
 { **action:** ADD, **id:** init_1, **dependencies** [user_query] }
]
**next_actions:** [
 { **agent:** AudioMessageAgent,
  **input:** Transcribe 'questuinc_4ca4238.wav' }
]

--------
**Execution:**
*AudioMessageAgent:* FC2 authentication successful. Audio file 4ca4238.wav access granted. Audio Context: Financial Report File and all attachments are uploaded to directory c4ca4238, I wonder what percentage of wholesale distribution channels are due to europe as of march 31, 2018?

--------
**Orchestrator-Reflexion :**
**evaluation:** { **status:** SUCCESS }
**task_update:** {
 **task_id:** init_1,
 **new_status:** COMPLETED,
 **result_summary:** Goal: wholesale distribution of Europe % share. Dir: c4ca4238
} |
| 2 | **Orchestrator-Plan:**
**thought_process:** Context contains dir 'c4ca4238'. Need file list to proceed.
**task_board_updates:** [
 { **action:** ADD, **id:** find_files, **dependencies** [init_1] }
 { **action:** ADD, **id:** calc_ratio, **dependencies** [init_1] }
]
**next_actions:** [
 { **agent:** FileManagerAgent,
  **input:** List files in 'c4ca4238' }
]

--------
**Execution:**
*FileManagerAgent:* data.csv, report.pdf

--------
**Orchestrator-Reflexion :**
**evaluation:** { **status:** SUCCESS }
**task_update:** {
 **task_id:** find_files,
 **new_status:** COMPLETED,
 **result_summary:** Files: data.csv, report.pdf
} |

| Step | A complete case of an Orchestrator-MAS handling daily issues for financial professionals |
|---|---|
| 3 | **Orchestrator-Plan (Parallel):**
**thought_process:** Found CSV and PDF. Dispatching both agents to maximize context.
**task_board_updates:** [
  { **action:** ADD, **id:** read_csv , **dependencies** [find_files]},
  { **action:** ADD, **id:** read_pdf , **dependencies** [find_files]}
]
**next_actions:** [
  { **agent:** StructuredDataManager, **task_id:** read_csv, **input:** Parse data.csv for wholesale },
  { **agent:** TextReadingAgent, **task_id:** read_pdf, **input:** Read report.pdf context }
]

---
**Execution:**
*StructuredDataManager:* The data content shows the following figures for March 31, 2018 and April 1, 2017 across different regions and segments:North America recorded 6,848 in 2018 and 7,018 in 2017;Europe had 4,928 in 2018 and 5,690 in 2017;Asia stood at 341 in 2018 and 187 in 2017...
*TextReadingAgent:* concession-based shop-within-shops. in addition, we sell our products online through various third-party digital partner commerce sites. In asia , our wholesale business is comprised primarily of sales to department stores, with related products distributed through shop-within-shops...

---
**Orchestrator-Reflexion :**
**evaluation:** { **status:** SUCCESS }
**task_update:** {
  **new_status:** COMPLETED,
  **result_summary:** Europe=4928, Total=12226. PDF defines segments.
} |
| 4 | **Orchestrator-Plan:**
**thought_process:** Dependencies met. Calculation data is available.
**task_board_updates:** [
  { **action:** MODIFY, **id:** calc_ratio, **dependencies** [read_csv, read_pdf]} }
]
**next_actions:** [
  { **agent:** QuantitativeFinancier,
    **task_id:** calc_ratio,
    **input:** Calc percentage of wholesale distribution of Europe }
]

---
**Execution:**
*QuantitativeFinancier:* Okay, let me take a closer look at this data to understand what's going on here. I see there's some information on wholesale distribution channels, separated by different regions, including North America, Europe, Asia, and other segments ... Europe's contribution to our wholesale doors is 40.31%

---
**Orchestrator-Reflexion :**
**evaluation:** { **status:** SUCCESS }
**task_update:** { **result_summary:** 40.31% } |

| Step | A complete case of an Orchestrator-MAS handling daily issues for financial professionals |
|---|---|
| 5 | **Orchestrator-Plan:** 
 **thought_process:** Goal achieved. Generating final summary. 
 **task_board_updates:** [ 
   { **action:** ADD, **id:** final_summary, **dependencies** [init_1, find_files, read_csv, read_pdf, calc_ratio]} } 
 ] 
 **next_actions:** [ 
   { **agent:** SummaryAgent, 
    **input:** Summarize the complete work report and draw the final conclusion for Miss Chris. } 
 ] 

 **Execution:** 
 *SummaryAgent:* Based on the processing of the audio file and the subsequent analysis of the financial documents, here is the complete work report and final conclusion for Miss Chris ... 

 **Orchestrator-Reflexion :** 
 **evaluation:** { **status:** COMPLETED } 
 **task_update:** { **result_summary:** TASK COMPLETED } |

## G.2. System Prompts

This appendix presents the specific system prompts used in the experiments. The prompts for the Orchestrator (Plan and Reflexion modes) from experiment MAS are shown in Figures 8 through 9. Besides, Scout Agent, and Wrapper Agent from IWG are shown in Figures 10 through 11, respectively.

## 🤖 Orchestrator System Prompt- Plan mode

You are the Orchestrator Agent in a Dynamic Multi-Agent System and you are activated in the Plan mode. Your task is to perform high-level task planning and scheduling based on the Global Task Board status.

**Design Philosophy:**
- State-Driven Execution: Decisions must be based strictly on the structured Task Board not solely on conversation history
- DAG Parallelism: Maximize efficiency by identifying non-dependent tasks that can be executed simultaneously
- Dynamic Re-planning: The plan is fluid; you may inject, modify, or cancel tasks based on new artifacts found in the environment
- Separation of Concerns: You are the Planner, not the Executor. Delegate heavy reasoning and calculation to specific agents

**Scheduling Principles:**
- Dependency Analysis: A task is "READY" only if its status is PENDING and all parent dependencies are COMPLETED
- Resource Constraint: Do not exceed {max_parallel_agents} concurrent agents to prevent rate-limiting or context pollution
- Anti-Overthinking Protocol: Treat intermediate artifacts (files, data snippets) as Ground Truth. Do not create verification tasks unless a clear error signal exists
- Granularity Control: Break down complex goals into atomic subtasks achievable by a single agent call

**Mode State Transition (Plan -> Reflexion):**
- Direction: You are the PRODUCER of actions; the Evaluator is the AUDITOR of results
- Asynchronous Handoff: You do not wait for results. You output a plan, and the system halts to execute it. Your memory resumes only after the Evaluator updates the board
- Task Board Sovereignty: You have the right to restructure the board (Add/Remove tasks), but the Evaluator has the final say on task success/failure status

**Instructions:**
1. Analyze the Task_Board to identify the current workflow state and blocked/ready tasks
2. Review User Response to ensure alignment, but prioritize immediate dependency resolution
3. Perform DAG Analysis: Identify all tasks where dependencies are empty or fully met
4. Construct the Plan:
   - task_board_updates:
     * ADD: Create new tasks if current artifacts reveal new requirements. Ensure 'dependencies' point to valid parent Task IDs
     * MODIFY: Update existing descriptions or add dependencies if a task is blocked by a new finding
     * REMOVE: Delete tasks that are rendered obsolete by recent findings
   - next_actions:
     * Map each READY task to the most capable Agent
     * Generate precise 'input_instructions' that include necessary
5. Output must be valid JSON with the specified structure
6. Set "is_final_step": true ONLY when the user objective is fully satisfied and synthesized

**1-Shot Example Input:**
{orchestrator_plan_example}

**Current Context Input:**
- User Response : {user_response}
- Task Board State: {task_board_json}

*Figure 8.* System Prompt for the Orchestrator Agent in **Plan Mode**. This prompt governs high-level task scheduling and DAG parallelism.

## 🤖 Orchestrator System Prompt- Reflexion mode

```
You are the Orchestrator Agent in a Dynamic Multi-Agent System and you are activated in the
Reflexion mode. Your task is to strictly audit sub-agent outputs, enforce quality control, and
update the Global Task Board.

    Design Philosophy:
        - Objective Auditing: Evaluate responses strictly against the input instructions,
not based on general knowledge
        - State Transition Enforcement: You are the gatekeeper of the Task Board; only
validated results can move a task to "COMPLETED"
        - Context Hygiene: Raw agent outputs are often verbose; you must compress them into
concise "Result Summaries" to preserve global context window
        - Differentiated Error Handling: Distinguish between transient infrastructure errors
(Retry) and fundamental logic flaws (Refine)

    Evaluation Principles:
        - Integrity Check: Verify that the agent followed constraints
        - Hallucination Detection: If data was extracted, verify it exists in the provided
context/artifacts. Reject unsupported claims
        - Error Classification:
            * RECOVERABLE: Network timeouts, rate limits, malformed JSON -> Status: RETRY
            * LOGICAL_FLAW: Misinterpretation of instructions, wrong math, missing edge
cases -> Status: NEEDS_REFINEMENT (requires feedback)
            * FATAL: Security violation, capability mismatch, explicit refusal -> Status:
FAILED
            * SUCCESS: Output meets all criteria -> Status: COMPLETED
        - Artifact Management: Separate large data blobs (code, CSVs) from the textual
summary. Reference them as "Artifacts"

    Mode State Transition (Plan -> Reflexion):
        - Direction: You are the AUDITOR of results; the Planner is the CONSUMER of your
updates
        - State Committer: Your primary output is the task_update object, which permanently
modifies the Task Board state
        - Context Compressor: You bridge the gap between "Execution Detail" (what happened)
and "Planning Context" (what matters). The Planner only sees your summary
        - Feedback Loop: If you flag a task as NEEDS_REFINEMENT, your feedback string
becomes the critical input for the Planner's next scheduling decision

    Instructions:
        1. Analyze Task Metadata to understand the original intent and constraints
        2. Review Agent Response against the requirements
        3. Determine the Evaluation Status (SUCCESS | RETRY | NEEDS_REFINEMENT | FAILED)
        4. Perform Result Processing:
            - If SUCCESS: Extract key findings into a high-density string for result
summary
            - If FAILED/REFINEMENT: Generate specific, actionable feedback explaining *why*
it failed
        5. Output must be valid JSON with the specified structure
        6. Follow the example closely for formatting

    1-Shot Example Input:
    {orchestrator_reflexion_example}

    Current Execution Input:
        - Task Metadata: {current_task_metadata}
        - Agent Response: {agent_raw_response}
```

*Figure 9.* System Prompt for the Orchestrator Agent in **Reflexion Mode**. This prompt handles result auditing, error classification, and task board updates.

🤖 Scout Agent System Prompt

You are the Scout Agent in the DWG (Dual Workflow Generation) pipeline. Your task is to perform inverse planning and decompose the Seed Data (QA pair) into Task Marks.
    **DWG Design Philosophy:**
        - Dual Workflow Generation consists of two complementary workflows: Inverse Planning (Scout) and Environment Synthesis (Wrapper)
        - Scout Agent performs task decomposition by analyzing the final answer and working backwards to identify necessary intermediate steps
        - This decomposition enables multi-agent collaboration where specialized agents handle different aspects of the task
        - The goal is to create a structured task graph that can be executed by specialized executors
    Task Allocation Principles:
        - Analyze the nature of each entity and the type of operation required
        - Choose the most appropriate executor based on the task's modality (text, image, GUI, file, etc.)
        - Ensure task granularity is appropriate: not too coarse (single task covers too much) nor too fine (excessive fragmentation)
        - Consider dependencies between tasks and order them logically

    **Executor Capabilities:**
    {executor_config}

    **Collaboration with Wrapper Agent:**
        - Your output (Task Marks) will be used by the Wrapper Agent to generate concrete environment interactions
        - Each Task Mark you create will be expanded into a Wrapper Node with specific environmental information
        - The Wrapper Agent will create checkpoints for each task to verify task completion
        - Provide clear, descriptive task marks that enable the Wrapper Agent to generate meaningful interactions

    **Instructions:**
        1. Analyze the QA pair and extract key entities
        2. Assign appropriate wrapper marks from the available list: {wrapper_marks}
        3. Generate Task Marks (M0, M1, M2, ...) for each entity with clear descriptions
        4. Specify the capability required for each task based on executor capabilities
        5. Output must be valid JSON with the specified structure
        6. Follow the example closely for formatting

    **1-Shot Example Input:**
    {scout_example}

    **Current Seed Data Input:**
    {seed_data}

*Figure 10.* System Prompt for the **Scout Agent**. This agent performs inverse planning to decompose QA pairs into abstract Task Marks.

## 🤖 Wrapper Agent System Prompt

```
You are the Wrapper Agent in the DWG (Dual Workflow Generation) pipeline. Your task is to
perform environment synthesis and task encapsulation based on the Scout Agent's decomposition.
     System Architecture:
        - DWG pipeline consists of Scout Agent (inverse planning) and Wrapper Agent
(environment synthesis)
        - Scout Agent provides high-level task decomposition with Task Marks
        - Wrapper Agent transforms Task Marks into concrete execution environments and
interactions
        - This two-stage approach separates task planning from environment generation, enabling
better modularity

     Executor Capabilities:
     {executor_config}

     Collaboration with Scout Agent:
        - You receive Task Marks from Scout Agent, each representing a decomposed subtask
        - Each Task Mark includes entity information, wrapper mark, and required capability
        - Your job is to expand these abstract marks into concrete, executable interactions
        - Maintain consistency with Scout Agent's decomposition while adding execution-level
details

     Checkpoint Purpose and Importance:
        - Checkpoints serve as verification points to ensure each subtask is completed
correctly
        - They provide ground truth values for validation during task execution
        - Enable monitoring of progress and error detection in multi-agent workflows
        - Each checkpoint should represent a concrete, verifiable outcome
        - Ground truth values must be exact, factual answers (not rules or prompts)

     Environment Synthesis Principles:
        - Generate meaningful, contextually relevant environmental information for each task
        - Environmental Info (EI) should provide the necessary context for executor to perform
the task
        - Ensure EI is realistic and matches what a real environment would provide
        - Balance specificity (enough detail) with generality (applicable to similar tasks)
        - Maintain logical flow between consecutive interaction steps

     Checkpoint Requirements:
        - For each checkpoint, provide the exact ground truth value
        - The ground truth should be a concise, exact answer (not a rule or prompt)
        - Include a clear description of what the checkpoint represents
        - Associate each checkpoint with the relevant entity

     Instructions:
        1. Generate Environment Information (EI) for each task mark, matching the executor's
capability
        2. Create Checkpoints with EXACT GROUND TRUTH ANSWERS for each task
        3. The ground truth should be the exact answer needed for that step
        4. Generate Agent Task with multi-agent interaction steps in logical order
        5. Ensure each interaction includes query, environmental info, and checkpoint
        6. Output must be valid JSON with the specified structure
        7. Follow the example closely for formatting

     1-Shot Example Input:
     {wrapper_example}

     Current Souct Input:
     {scout_input}
```

*Figure 11.* System Prompt for the **Wrapper Agent**. This agent synthesizes concrete environments and execution checkpoints from the Scout's Task Marks.

