# OpenReview forum: "Recognize Your Orchestrator: An Entropy Dynamics Perspective for LLM Multi-Agent Systems"
_ICML.cc/2026/Conference — ICML 2026 regular_

### Official Review · Reviewer_y34U · 2026-02-27

**Soundness:** 3
**Presentation:** 3
**Significance:** 3
**Originality:** 2
**Overall Recommendation:** 3
**Confidence:** 5

**Summary:**

The paper investigates the stability and performance of Large Language Model Multi-Agent Systems (LLM MAS) through the lens of physics-inspired entropy dynamics. The authors identify a phenomenon termed the "Reasoning Trap," where high-reasoning models (e.g., o1-preview) perform exceptionally in individual tasks but exhibit systemic collapse when serving as a central orchestrator in complex workflows. To analyze this, they develop a Mean-Field Entropy Dynamics framework that models the evolution of task uncertainty (entropy) against cumulative context load. Furthermore, the paper introduces Inverse Workflow Generation (IWG), a method to create high-complexity benchmarks with dense, verifiable checkpoints. The core contribution is the identification of a critical "phase transition" where context-induced noise overrides a model's resolution capacity, leading to a sudden divergence in system entropy and task failure.

**Compliance With Llm Reviewing Policy:**

Affirmed.

**Final Justification:**

The paper presents an interesting and novel perspective on LLM-based multi-agent systems by modeling orchestrator behavior through entropy dynamics. The strengths of the work lie in its cross-disciplinary framing, the identification of the “Reasoning Trap,” and the introduction of IWG as a tool for constructing step-verifiable workflows. The paper is also clearly written and provides a coherent narrative connecting theory and empirical observations.

In terms of soundness, the rebuttal has improved my confidence of the work. However, several core concerns remain. The proposed mean-field entropy dynamics formulation still relies on simplifying assumptions that may not hold in more complex or decentralized MAS settings. While the authors clarified the role of IWG, questions about the realism and generalizability of the synthetic task construction persist. Moreover, despite the additional experiments, the overall empirical evaluation remains relatively limited in scale and diversity compared to real-world large-scale MAS systems. The theoretical formulation also remains somewhat phenomenological, with limited grounding in the underlying decision processes.

Overall, while the rebuttal improves clarity and partially strengthens the empirical evidence, it does not fully address concerns regarding generality, scalability, and theoretical grounding. Therefore, my overall assessment remains largely unchanged, and I maintain my original recommendation.

**Key Questions For Authors:**

1.**Defining $H(t)$**: Could you provide the specific mathematical formulation used to calculate the entropy $H$ from the experimental MAS logs? Is it a heuristic based on checkpoint completion or a probabilistic measure of the model's output?

2.**Decentralized MAS**: How would the Entropy Dynamics framework adapt to decentralized architectures where there is no single orchestrator? Does the "system entropy" still follow a mean-field trajectory?

3.**Noise Mitigation**: Given the high $\beta$ (noise sensitivity) found in reasoning models, did you test if specific techniques like "Context Compression" or "Memory Summarization" shifted the $C_{crit}$ point significantly?

4.**Parameter Stability**: How stable are the $\alpha$ and $\beta$ parameters across different task domains? For example, is a model's resolution power in coding tasks predictive of its power in creative writing MAS?

5.**Verification of the IWG Data Fidelity**:
The Inverse Workflow Generation (IWG) explicitly constructs reasoning chains backward from the target result. Does this backward construction inherently introduce a "simplicity bias" where the generated paths are more logically linear than real-world forward-execution tasks? I would like the authors to discuss how they ensured that IWG-synthesized trajectories retain the "messy" causal dependencies found in spontaneous human-agent interactions. This response is critical to my confidence in the Rigorousness of your empirical validation.

6.**The "Reasoning Trap" vs. Token Efficiency**:
You argue that heavy-thinking models fail due to "Context Squeezing" caused by internal CoT. However, is it possible that the failure is due to a lack of "instruction following" rather than purely an "attention budget" issue? Since you introduced the "Token Efficiency" metric, could you provide data on whether forcing a reasoning-heavy model to use a very large context window (e.g., 128k+) mitigates the failure? If the performance does not improve with a larger window, it would suggest the "Reasoning Trap" is a logical coherence issue rather than a physical "squeezing" issue, which would change the Originality of your interpretation.

**Limitations:**

Yes. The authors adequately discuss the limitations regarding the mean-field approximation and the potential for "over-fitting" a specific MAS architecture. They also address the societal impact of MAS reliability in critical decision-making.

**Strengths And Weaknesses:**

**Strengths**

**Originality through Cross-Disciplinary Insights**: The work provides a highly novel perspective by applying Mean-Field Entropy Dynamics to LLM Multi-Agent Systems (MAS). This approach moves beyond simple performance metrics to offer new insights into the "process dynamics" of how agentic systems fail.

**Identification of the "Reasoning Trap"**: The paper identifies and theoretically explains a critical property of high-reasoning models: while they possess high resolution power, they are also more sensitive to noise in orchestrator roles. This highlights an important and previously under-explored property of existing SOTA models.

**Innovative Benchmarking (IWG)**: The introduction of Inverse Workflow Generation (IWG) is a significant methodological contribution. It allows for the creation of high-complexity tasks with dense, verifiable checkpoints, which advances the field's ability to evaluate MAS reliability beyond simple end-result metrics.

**Strong Narrative and Structure**: The presentation is excellent, with a clearly written and well-structured argument. The overall narrative is easy to follow, successfully positioning the work against current MAS literature while clearly discussing its unique contributions.

**Weaknesses**

**Simplification via Mean-Field Assumption**: While the theoretical results are technically sound within their own framework, the Mean-Field Assumption—treating agent interactions as a uniform field—is a restrictive assumption. It may fail to capture highly non-linear or recursive conflicts between agents in real-world scenarios.

**Imbalance between Theory and Broad Empirical Validation**: The claims regarding noise sensitivity ($\beta$) and resolution capacity ($\alpha$) are well-supported by the experiments provided, but the evaluation is limited to a few top-tier models. The paper would benefit from a broader empirical study across more model families (e.g., Llama or Qwen) to prove the "Reasoning Trap" is a universal phenomenon.

**Heuristic Nature of Entropy Calculation**: Although the methods used are appropriate, the specific calculation of "Entropy" ($H$) relies on specific MAS logs and checkpoint completions. This could be seen as a domain-specific improvement that may require clearer justification to ensure the results are fully reproducible by other expert readers.

**Architecture Scope**: The significance of the results is primarily demonstrated for centralized orchestrator-worker architectures. The paper does not clearly discuss how these dynamics differ in decentralized or swarm-based MAS, which slightly limits the broad applicability of the current findings.

---

> ### Author Rebuttal · Authors · 2026-03-31
>
> Thank you for your insightful comments. Our response is as follows：
>
> ### **Response to Question 1 and Weakness 3**
>
> H(t) is not a heuristic based on checkpoint completion. It is the Shannon entropy, defined in paper Eq. (1),  of the Orchestrator’s executor-routing distribution. In experiments, $p_i$ is estimated from MAS logs by aggregating executor selections at each step across trajectories. Thus, entropy is computed from routing behavior, while checkpoints are used only for control and evaluation, not for defining entropy. The framework is therefore not specific to IWG; any MAS with step-level orchestrator monitoring can be analyzed in the same way.
>
> ---
>
> ### **Response to Question 2 and Weakness 4**
>
> We would like to clarify that our research focuses on mainstream practical MAS, the centralized orchestration paradigm. The current mean-field entropy model could not be directly generalized to fully decentralized MAS.
> For decentralized settings, a feasible extension is a combination of local policy entropies and communication entropy over the interaction graph.
>
> ---
>
> ### **Response to Question 3**
>
> We added an ablation comparing full-history with context compression under the same orchestrator. Compression consistently improves orchestration quality, increasing step-level sr at almost all stages ( Step-SR at steps 2–4 rises from 0.27 / 0.12 / 0.04 to 0.38 / 0.19 / 0.17) while reducing entropy.
>
> Fitted dynamics further show that compression reduces both β (0.46 → 0.32) and A_task (3.16 → 2.54), slowing entropy growth and delaying instability. The shift is not dramatic because the exploration pattern changes only mildly ω(1.72 → 1.63). In short, compression mitigates noise accumulation, but system entropy remains jointly shaped by task structure and model behavior.
>
> ---
>
> ### **Response to Question 4**
>
> Our results do not support the strong claim that the fitted parameters are domain-invariant, The parameters in the paper are not universal model fingerprints. To examine this more directly, we fitted two representative domains:
>
> | Domain          | A_task |    γ |    ω |    β |
> | --------------- | ------ | ---- | ---- | ---- |
> | File Operation  |   2.53 | 1.02 | 1.63 | 0.32 |
> | MCP/API Calling |   0.49 | 0.46 | 2.93 | 0.01 |
> | Overall         |   3.16 | 1.56 | 1.72 | 0.47 |
>
> These results show clear cross-domain variation, consistent with differences in context length, rule structure, tool feedback, and interaction noise. For example, API-use tasks typically provide more structured and concise contexts, which is consistent with the much smaller fitted β.
>
> ---
>
> ### **Response to Question 5 and Weakness 1**
>
> Thank you for your important question. We respectfully clarify that IWG does not synthesize execution trajectories. It synthesizes the task and environment; the actual trajectory is still generated by the MAS through forward execution. The question, therefore, is not whether we hard-code a linear script, but whether the synthesized task structure is overly simplified.
>
> On this point, IWG is explicitly formulated as a DAG workflow, not a single linear chain. As our response in (Reviewer qJS6)-Question 1, the resulting tasks remain causally complex rather than collapsing into an artificially clean regime. We therefore view IWG as preserving substantial causal structure while prioritizing process verifiability over unconstrained real-world messiness.
>
> ---
>
> ### **Response to Question 6**
>
> The observed failure is not primarily a breakdown of instruction following. Our log analysis shows that even in late-stage failures, the Orchestrator usually still preserves the required structured communication and task-board protocol. What degrades is the higher-level control layer: dependency tracking, global state management, and sustained convergence over long horizons.
>
> We agree that a same-model larger-context-window test would be ideal. However, switching to another long-context variant would introduce architecture and alignment confounds, so we will state this limitation explicitly rather than overclaiming causal isolation.
>
> ---
>
> ### **Response to Weakness 2**
>
> We additionally examined Llama-4-Maverick. Since the Llama family does not provide a native thinking mode, we used a two-stage prompt reasoning setup. The fitted results are:
>
> | Model            | A_task |     γ |     ω |     β |
> | ---------------- | ------ | ----- | ----- | ----- |
> | Llama-4-Maverick |  0.632 | 0.791 | 1.954 | 1.561 |
>
> This suggests that the entropy-dynamics framework also extends to Llama-family models. At the same time, for relatively weaker models with limited reasoning depth, moderate guided reasoning can still be beneficial rather than always inducing a strong “Reasoning Trap.” We will clarify this boundary and avoid overstating universality. Besides, the Qwen family is already covered in the main text, with Table 3 and Figure 4.

---

> > ### Author Rebuttal · Reviewer_y34U · 2026-04-02
> >
> > Thank you for the detailed and thoughtful rebuttal. The authors have addressed several of my questions with additional clarifications and empirical analyses.
> >
> > However, some of my core concerns remain only partially addressed. The mean-field formulation still relies on simplifying assumptions that may limit its applicability in more complex or decentralized multi-agent systems. While the authors clarified the role of IWG, questions regarding the realism and generalizability of the synthesized task environments remain. Additionally, although the empirical evaluation has been expanded, it is still relatively limited in scale and diversity compared to real-world MAS deployments.
> >
> > Overall, the rebuttal improves clarity and strengthens the empirical support, but does not fully resolve concerns regarding generality, scalability, and theoretical grounding.

---

> > > ### Author Response · Authors · 2026-04-07
> > >
> > > We sincerely thank you for the constructive dialogue. We appreciate the recognition that our previous rebuttal "improves clarity and strengthens empirical support." Regarding the remaining concerns, we offer the following clarifications:
> > >
> > > # 1. On the "Mean-Field" Simplification and Decentralized MAS
> > >
> > > We respectfully clarify that the mean-field approximation is a deliberate modeling choice intended to extract macroscopic governing laws from the high-dimensional, stochastic token-level interactions of an Orchestrator. It is not a simplifying assumption that the interactions inside the multi-agent system are simple; instead, it is a way to describe the system at the collective level. An analogy is from thermodynamics: when we describe a gas by its temperature or pressure, we do not assume that each molecule moves in a simple way, nor do we track every collision one by one. Our mean-field model plays a similar role; in this sense, the entropy curve is not meant to replace the full microscopic process, but to provide a compact description of its behavior. The key point is that, when many scheduling events accumulate, the overall trajectory of this uncertainty becomes regular and informative, even though the individual decisions remain complex. Experimental results and analysis show that the mean-field model identifies the "Reasoning Trap" and the context-induced "Entropy Saturation" point that would be obscured by microscopic noise in raw logs.
> > >
> > > Further, to validate our framework in decentralized settings, we implemented a decentralized MAS inspired by AgentNet[1]. Specifically, we removed the central controller, broadcasted tasks to all agents, and allowed each agent to execute distributively within its own workspace, independently deciding which neighbors to communicate with and which executor or downstream node to schedule. In this decentralized system, each node can be viewed as a local instance of the orchestrator (as discussed in the last rebuttal), and the overall system entropy can be analyzed as an aggregation (we use average aggregation) of individual local entropy trajectories. In this way, our mean-field model is extended to a decentralized MAS. The experimental results are as follows:
> > >
> > > | Metric       | A_Task | γ    | ω    | β    |
> > > |--------------|--------|------|------|------|
> > > | decentralized | 8.57   | 0.92 | 0.67 | 0.95 |
> > > | centralized   | 3.16   | 1.56 | 1.72 | 0.47 |
> > >
> > > The result shows in the decentralized system, A_Task has a bigger value of 8.57 than 3.16 of centralized MAS system. This indicates a significant rise in task pressure per node of the decentralized system. Moreover, with full task loading and more frequent communication, the contextual burden increases correspondingly (β: 0.47 → 0.95), while the speed of task progression and exploration (γ, ω) falls short of centralized MAS.
> > >
> > > # 2. On IWG and Extension to Real-world Environments
> > > We build IWG to provide a scientific and fair synthesized environment to facilitate community research. To address your concerns about realism of synthesized environment and our mean-field formulation in real-world environments, we conducted an additional experiment in a real-world MAS environment. It contains:
> > > - the real-world web environment on April 5, 2026, including live online queries, real-time data fetching, and time-sensitive information retrieval;
> > > - Magentic-One system[2], a representative real-world multi-agent system developed by Microsoft;
> > > - GPT-4o as the common base model and GAIA[3] as the task input.
> > >
> > > The experimental results are as follows:
> > >
> > > | Metric                | A_Task | γ    | ω    | β    |
> > > |-----------------------|--------|------|------|------|
> > > | Real-world  | 6.42   | 1.14 | 1.03 | 0.68 |
> > > | IWG                   | 3.16   | 1.56 | 1.72 | 0.47 |
> > >
> > > Successful fitting of mean-field formulation parameters in real-world environments shows that it can be used to analyze the behaviors of real-world MAS deployments. A bigger A_Task value may be due to noises (such as network latency, API instability) in real-world environments triggering the context management mechanism of MAS, which converts these challenges into filtering noise and producing intermediate summaries. Overall, there are similar fitting values of real-world environments and IWG, which indicates that our IWG approximates the real-world environment.
> > >
> > > The two supplementary experiments above demonstrate the applicability of our framework to decentralized MAS and real-world environments, as well as the realism of the IWG synthesized environment. We sincerely hope they could address your concerns.
> > >
> > > [1] Yang, Yingxuan, et al. "AgentNet: Decentralized Evolutionary Coordination for LLM-based Multi-Agent Systems." NeurIPS 2025.
> > >
> > > [2] Microsoft Research AI Frontiers. "Magentic-One: A Generalist Multi-Agent System for Solving Complex Tasks." arXiv preprint arXiv:2411.04468.
> > >
> > > [3] Mialon, Grégoire, et al. "GAIA: a benchmark for General AI Assistants." ICLR. 2024.

---

### Official Review · Reviewer_jWP1 · 2026-03-05

**Soundness:** 4
**Presentation:** 4
**Significance:** 4
**Originality:** 4
**Overall Recommendation:** 6
**Confidence:** 3

**Summary:**

This paper explores the "Orchestrator Bottleneck" within Multi-Agent Systems (MAS).  The authors conducted a failure attribution analysis and revealed that centralized managers are responsible for over 67% of task failures.  To analyze this fragility, the authors propose a Mean-Filed Entropy Dynamics framework that models orchestration as a physical system balancing task resolution against the growing pressure of context accumulation.  In addition, due to the lack of high-density, trajectory data, they introduce Inverse Workflow Generation, a method for creating high-complexity benchmarks by synthesizing reasoning chains backward to ensure verifiable intermediate steps.  A notable insight from the work is the "Reasoning Trap", where models with heavy internal thinking struggle as orchestrators because their self-generated dialogue squeezes the attention budget and dilutes focus.   It recommends adopting instant breadth-first thinking to prevent the model from hallucinating on unnecessary complexity to enhance orchestration.

**Compliance With Llm Reviewing Policy:**

Affirmed.

**Key Questions For Authors:**

None

**Limitations:**

Yes

**Strengths And Weaknesses:**

Strengths:
Every aspect of the research was conducted methodically with rigor, from the initial failure attribution analysis, to the theoretical framework development, to the Inverse workflow generation and the metrics used for evaluating orchestrator performance.  The submission is clearly written and well structured, key insights are delivered in the main body and additional details can be found in the appendix.  This work is highly significant in that it provides a deep analysis in the observed fragility of MAS system and identified a counter-intuitive phenomenon, the reasoning trap, that gives insights into how to enhance task orchestration.  The novelty of the mean-field entropy dynamics model and the inverse workflow generation pipeline and the new insight into context squeezing highlight the work's originality.
Weaknesses:
The analysis of entropy dynamics is bounded to a 6-step orchestration window.  There may be opportunity for future research to extend similar analysis to ultra-long horizon tasks.

---

> ### Author Rebuttal · Authors · 2026-03-31
>
> We sincerely thank you for the highly positive assessment and strong support. We are glad that you recognized the originality of the mean-field entropy dynamics framework and IWG pipeline, and the significance of the “Reasoning Trap” insight for MAS orchestration.

---

> > ### Author Rebuttal · Reviewer_jWP1 · 2026-03-31
> >
> > Nothing to be resolved.

---

### Official Review · Reviewer_qJS6 · 2026-03-06

**Soundness:** 3
**Presentation:** 3
**Significance:** 3
**Originality:** 3
**Overall Recommendation:** 5
**Confidence:** 3

**Summary:**

This work examines the orchestrator bottleneck inherent to centralised multi-agent systems (MAS). It reveals that system collapses are largely driven by the central orchestrator struggling with growing context and task complexity, not by the limitations of individual sub-agents. The researchers mathematically ground this issue using a Mean-Field Entropy Dynamics framework that treats orchestration choices as a continuous probability field. To test this theoretically, an Inverse Workflow Generation (IWG) pipeline is introduced, creating benchmark data via backwards-chaining from established solutions to yield verifiable intermediate checkpoints. Testing uncovers a critical "Reasoning Trap": models heavily reliant on reasoning thrive in siloed tasks, but their voluminous chain-of-thought outputs cause "context squeezing." Consequently, these models fail catastrophically when forced to serve as orchestrators.

**Compliance With Llm Reviewing Policy:**

Affirmed.

**Final Justification:**

I would like to thank the authors for their detailed and highly effective rebuttal. They have addressed my questions and concerns point by point. Therefore, I am raising my score to 5.

**Key Questions For Authors:**

I would be happy to raise my score if the authors can address the following questions:

1. The Inverse Workflow Generation (IWG) uses a Wrapper agent to synthesise the environment, e.g., generating mock API or terminal outputs, rather than using a real execution engine. Do these synthesised environments adequately capture the unpredictable, out-of-distribution noise of real-world tool use?

2. There is a growing body of recent literature investigating how different factors impact the effectiveness of LLM MAS [1-3].  Could you better contextualise your contributions in relation to this specific recent literature? How does your Entropy Dynamics perspective explain or contrast with their findings?

[1] Yang, Yongjin, et al. "Revisiting multi-agent debate as test-time scaling: A systematic study of conditional effectiveness." arXiv preprint arXiv:2505.22960 (2025).

[2] Gao, Mingyan, et al. "Single-agent or Multi-agent Systems? Why Not Both?." arXiv preprint arXiv:2505.18286 (2025).

[3] Tang, Bohan, et al. "On the Importance of Task Complexity in Evaluating LLM-Based Multi-Agent Systems." Workshop on Scaling Environments for Agents.

**Limitations:**

yes

**Strengths And Weaknesses:**

Strengths:

1. The paper is generally well-written, logically structured, and easy to follow.

2. The empirical observation of the Reasoning Trap is a timely finding. The ablation studies comparing models with and without heavy reasoning provide compelling evidence for the hypothesis. Furthermore, the IWG data generation pipeline is methodologically sound, employing a rigorous validation including a human-in-the-loop expert review to ensure trajectory quality and solvability.

3. The paper addresses a critical bottleneck in MAS. The finding regarding the Reasoning Trap is valuable given the current industry trend toward scaling inference-time compute. Proposing that orchestrators require a different inductive bias provides practical utility for people designing MAS.

Weaknesses:

1. The IWG pipeline uses a Wrapper agent to mock the environment; evaluating models on synthesised text environments rather than real interactive environments might introduce a gap.

2. The literature review is somewhat incomplete. It omits some relevant recent studies from 2025 [1-3] that explore how different factors impact the effectiveness of LLM MAS. Properly contextualising the proposed framework against these works would strengthen the paper.

[1] Yang, Yongjin, et al. "Revisiting multi-agent debate as test-time scaling: A systematic study of conditional effectiveness." arXiv preprint arXiv:2505.22960 (2025).

[2] Gao, Mingyan, et al. "Single-agent or Multi-agent Systems? Why Not Both?." arXiv preprint arXiv:2505.18286 (2025).

[3] Tang, Bohan, et al. "On the Importance of Task Complexity in Evaluating LLM-Based Multi-Agent Systems." Workshop on Scaling Environments for Agents.

---

> ### Author Rebuttal · Authors · 2026-03-31
>
> Thanks for your helpful suggestion and important concern. Our response is as follows：
>
> ---
>
> ### **Response to Question 1**
>
> Actually, the synthesized environment preserves enough uncertainty, disturbance, and recovery burden to realistically stress the Orchestrator. In our benchmark design, we do not rely on purely clean synthetic data only: as described in the Appendix D, we explicitly include exception and recovery paths derived from realistic MAS workflows, and inject such noisy events into IWG instances in a controlled manner. This is also exactly what EH-F1 measures, by comparing the model’s recovery action. To directly evaluate this, we compared Golden Context, Exception Injection, and Raw Environment settings, The results are shown below.
>
>
> | Step-SR  |    1 |    2 |    3 |    4 |    5 |    6 |
> | ------------------- | ---- | ---- | ---- | ---- | ---- | ---- |
> | Golden Context      | 0.96 | 0.40 | 0.17 | 0.07 | 0.08 | 0.16 |
> | Exception Injection | 0.92 | 0.27 | 0.12 | 0.04 | 0.09 | 0.19 |
> | Raw Environment     | 0.94 | 0.50 | 0.26 | 0.16 | 0.18 | 0.17 |
>
>
> | Entropy  |    1 |    2 |    3 |    4 |    5 |    6 |
> | ------------------- | ---- | ---- | ---- | ---- | ---- | ---- |
> | Golden Context      | 2.78 | 3.28 | 3.15 | 3.35 | 2.72 | 2.91 |
> | Exception Injection | 3.03 | 3.39 | 3.02 | 3.20 | 2.55 | 2.53 |
> | Raw Environment     | 2.44 | 2.93 | 2.86 | 2.97 | 2.62 | 3.07 |
>
>
> | Parameters | A_task | γ    | ω    | β    |
> |-----------------------|--------|------|------|------|
> | Golden Context        | 0.49   | 0.10 | 1.45 | 0.16 |
> | Exception Injection   | 9.85   | 1.20 | 1.23 | 0.32 |
> | Raw Environment       | 3.16   | 1.56 | 1.72 | 0.47 |
>
>
> These results show that introducing noisy environments does not collapse the workflow into a different regime, and the entropy trajectory preserves a stable overall shape and transition trend across clean, noisy, and raw settings. Therefore, while IWG uses synthesized environments for achieving step-wise controllability, it still captures the exception burden and noise sensitivity that characterize real-world MAS tool use, and our entropy-dynamics framework remains valid under such noisy workflows.
>
> ---
>
> ### **Response to Question 2**
>
> Thank you for your helpful advice, we will discuss the three papers in the revised version. Yang et al. study multi-agent debate as a form of test-time scaling and show that its gains are conditional on factors such as task difficulty and model capability, while agent diversity brings limited benefit in some settings. Gao et al. compare single-agent and multi-agent systems directly, showing that MAS advantages can shrink as frontier models improve, and propose a hybrid cascading design between SAS and MAS. Tang et al. argue that task complexity is central for evaluating LLM-MAS, and characterize it through reasoning depth and capability width.
>
> These works are complementary to ours rather than redundant. In contrast to debate-style MAS or SAS-vs-MAS comparisons, our paper isolates the central orchestration layer and asks a different question: why does an orchestration MAS fail once complexity and context accumulation grow? This is precisely the gap our paper addresses through the mean-field entropy framework. In our setting, we hold executors fixed and analyze how uncertainty evolves inside the orchestrator, showing that the orchestrator's significant vulnerability accounts for most failures and that performance collapse is closely tied to context loading and entropy growth.
>
> In the revision, we will update Sections 1 and 5 to make this positioning explicit: Yang et al. motivate the need to study conditional MAS gains, Gao et al. highlight the capability-cost tradeoff and hybrid design space, and Tang et al. support our emphasis on task complexity. We will also clarify that our contribution is mechanism-level: rather than only identifying when MAS helps, we model how centralized orchestration destabilizes under growing history, and provide step-level evidence through entropy fitting.

---

> > ### Author Rebuttal · Reviewer_qJS6 · 2026-04-02
> >
> > I would like to thank the authors for their detailed and highly effective rebuttal. You have addressed my questions and concerns point by point. Therefore, I am raising my score to 5. Please ensure that all the details provided in this rebuttal are incorporated into the final camera-ready version of the paper as promised.

---

### Official Review · Reviewer_fYUP · 2026-03-12

**Soundness:** 2
**Presentation:** 2
**Significance:** 1
**Originality:** 2
**Overall Recommendation:** 3
**Confidence:** 4

**Summary:**

The paper studies the role of the orchestrator in LLM-based multi-agent systems and argues that many system failures originate from the orchestration layer rather than from individual executor agents. To analyze this behavior, the authors propose an entropy-based formulation that measures the uncertainty of the orchestrator’s agent-selection decisions at each step. They introduce a mean-field entropy dynamics equation intended to capture how this uncertainty evolves as a result of two competing effects: task resolution, which reduces uncertainty, and context accumulation, which increases it. To obtain step-level data for analysis, the paper also proposes an Inverse Workflow Generation (IWG) procedure that constructs synthetic multi-agent tasks with intermediate checkpoints so that step-wise behavior can be evaluated. Experiments across several LLMs report orchestration benchmarks and fit the proposed entropy dynamics model to observed entropy trajectories.

**Compliance With Llm Reviewing Policy:**

Affirmed.

**Final Justification:**

The authors’ rebuttal provides useful clarifications and strengthens the empirical evidence, particularly in supporting the relationship between entropy and step-level correctness.

However, I still find that the core methodological contribution remains insufficiently grounded, as the entropy dynamics formulation appears more descriptive than principled.

In addition, I am not yet fully convinced that the conclusions drawn from synthesized environments will generalize to real-world multi-agent systems.

**Key Questions For Authors:**

- The paper observes that step accuracy decreases as the number of steps increases and interprets this behavior through the lens of entropy dynamics. However, such a trend may also arise naturally from error propagation in multi-step reasoning processes. Could the authors clarify how they disentangle the effect of entropy dynamics from standard error accumulation? Are there ablation studies or controlled experiments that isolate these factors?

- The paper introduces Inverse Workflow Generation (IWG) to construct step-verifiable workflows for analysis. However, step-level trajectories could in principle also be obtained by logging execution traces from existing MAS frameworks on real tasks. Could the authors clarify why the inverse construction is necessary for studying entropy dynamics, and whether similar conclusions would hold when analyzing real MAS execution logs?

- The paper suggests that entropy dynamics are related to step-level correctness, yet the experimental analysis primarily presents qualitative trends. Could the authors provide quantitative evidence (e.g., correlation analysis, regression, or predictive evaluation) demonstrating that entropy is meaningfully associated with or predictive of step-level success?

- The proposed entropy dynamics model is evaluated on workflows with relatively short horizons (approximately six steps). In practical multi-agent systems, however, tasks may involve significantly longer reasoning chains and more complex dependency structures. Could the authors discuss whether the proposed entropy dynamics model is expected to generalize to long-horizon MAS reasoning, and whether the observed dynamics remain stable as the workflow length increases?

**Limitations:**

yes

**Strengths And Weaknesses:**

**Strengths**

- **Originality**: The paper introduces an entropy-based perspective for analyzing the behavior of LLM orchestrators in multi-agent systems, framing orchestration as a dynamical process rather than evaluating only end-task performance. This conceptual framing is relatively uncommon in current MAS literature and provides a new lens for studying coordination behavior.
- **Significance**: The work focuses on the orchestration layer of multi-agent systems, an increasingly important component in agentic architectures. By highlighting the orchestrator as a potential bottleneck and providing empirical comparisons across a range of LLMs, the study draws attention to a system-level challenge that is relevant to the design of future agent frameworks.
- **Soundness**: The paper attempts to combine empirical measurements with a simplified dynamical model, fitting entropy trajectories observed from multi-agent executions. This effort to connect observable system behavior with interpretable parameters represents a reasonable first step toward quantitative analysis of orchestrator dynamics.

**Weaknesses**:
- **Presentation**: The narrative structure of the paper is at times difficult to follow, particularly regarding the relationship between the entropy dynamics formulation (Section 2) and the Inverse Workflow Generation (Section 3). The role of IWG in validating the proposed model is not clearly articulated, and the transition between theoretical modeling and dataset construction appears abrupt, making it challenging for readers to understand how these components are tightly connected.
- **Soundness**: The proposed entropy dynamics equation appears largely phenomenological. The functional form is introduced without a rigorous derivation from the underlying decision process of the orchestrator, and the physical interpretations assigned to parameters such as $A_{task}$, $\gamma$, $\omega$, and $\beta$ seem to be post-hoc explanations rather than theoretically grounded quantities.
- **Soundness**: The empirical analysis does not establish a clear statistical relationship between entropy values and step correctness. While the paper visually compares entropy trajectories and accuracy trends, it does not provide correlation analysis, regression modeling, or other statistical tests that would substantiate the claim that entropy meaningfully predicts step-level correctness.
- **Soundness**: The entropy dynamics model is fitted using a very small number of time steps (approximately six), while the equation contains multiple free parameters. This raises concerns about potential overfitting, especially since the paper does not include simpler baseline models (e.g., linear or logarithmic fits) to demonstrate that the proposed formulation is necessary or superior.
- **Soundness**: The entropy measure is defined solely based on the probability distribution over agent selections. In many practical multi-agent systems, however, agent execution follows structured workflows or dependency graphs where the next action is constrained. As a result, the entropy metric may not accurately capture the true uncertainty of orchestration decisions in realistic systems.
- **Significance**: The necessity of Inverse Workflow Generation is not entirely convincing. Step-level trajectories can in principle be obtained by logging the execution traces of existing MAS frameworks, and it remains unclear why the inverse construction of workflows is required to study entropy dynamics.
- **Soundness**: The evaluation relies primarily on synthetic tasks generated by IWG rather than logs from real multi-agent deployments. Because these workflows are constructed by reversing from answers to reasoning steps, it is unclear whether the observed entropy dynamics reflect realistic orchestration behavior.

---

> ### Author Rebuttal · Authors · 2026-03-31
>
> Thank you for your insightful comments. Our response is as follows：
>
> ---
>
> ### **Response to Question 1: Error Propagation Ablation**
>
> We clarify a key conceptual distinction. Error propagation is an exogenous downstream effect: incorrect outputs from earlier steps corrupt later inputs. Entropy dynamics, by contrast, describes the endogenous evolution of the orchestrator’s decision uncertainty.
>
> To isolate these two factors, we performed a controlled ablation with a Golden Context setting, where each step receives full ground-truth results from all previous steps, completely eliminating error propagation, and compared it with the original Raw Environment.
>
> | Setting         | A_task |    γ |    ω |    β |
> | --------------- | ------ | ---- | ---- | ---- |
> | Golden Context  |   0.49 | 0.10 | 1.45 | 0.16 |
> | Raw Environment |   3.16 | 1.56 | 1.72 | 0.47 |
>
> Even when error propagation is blocked, the evolution trend and parameter relationship of entropy model remains. This shows that our entropy model captures an intrinsic property of orchestrator dynamics rather than merely downstream error accumulation.
>
> ---
>
> ### **Response to Question 2 and Weaknesses 1, 6, 7: Mechanism Clarification of IWG**
>
> Thank you for this valuable question. We respectively clarify that IWG does not synthesize MAS execution trajectories. It only constructs the task objective, execution environment, and deterministic step-level checkpoint rules. All trajectories in our experiments are produced by the MAS orchestrator through forward execution.
>
> This distinction is essential because our system entropy is computed from the orchestrator’s step-level executor scheduling distribution. To make this entropy meaningful, each scheduling step must correspond to valid task progression. In real commercial scenarios, we cannot access their complete execution environment, failing to evaluate step-level results.  In contrast, IWG provides a controlled but executable environment that guarantees task solvability, which supports our entropy dynamics analysis of MAS behaviors.
>
> ---
>
> ### **Response to Question 3 and Weaknesses 3, 4: Quantitative Evidence and Statistical Test**
>
> To provide quantitative evidence that entropy is meaningfully related to step-level correctness, we conducted three analyses on full step-level logs from all MAS tasks.
>
> Correlation analysis. The Pearson correlation between empirical entropy and step-level success rate is -0.82 with p < 0.01, indicating a strong and statistically significant negative association.
>
> Regression analysis. A univariate linear regression using step-level SR as the dependent variable and entropy as the only predictor achieves R² = 0.78.
>
> Predictive evaluation. We compared our entropy-based model against linear, 4th-order and 5th-order polynomial baselines on 12-step tasks.
>
> | Model        |    MSE |    R² | Val MSE |
> | ------------ | ------ | ----- | ------- |
> | Linear       |  0.079 | 0.050 |   0.439 |
> | Polynomial-4 |  0.018 | 0.897 |   0.517 |
> | Polynomial-5 | 0.013 | 0.902 |   2.582 |
> | Ours         |  0.019 | 0.782 |   0.242 |
>
> The 5th-order polynomial severely overfits, and although the 4th-order polynomial fits training data well, it generalizes worse. Our entropy-based model achieves the best balance between fit and generalization, supporting that entropy is not just descriptively correlated with step success, but also predictive of it.
>
> ---
>
> ### **Response to Question 4: Long-Horizon Task Analysis**
>
> We tested long-horizon generalization using 12-step tracking under a memory-compression condition. The fitted parameters (A_task=0.89, γ=0.09, ω=0.50,β=0.49) remain consistent with our theoretical framework, and entropy still shows a significant negative correlation with step-level success (Pearson r = -0.73, p < 0.05). This indicates that the core entropy-success relationship persists beyond the 6-step main setting.
>
> Besides, the 6-step design in the main paper was chosen because it aligns with common MAS benchmarks such as GAIA and SkillsBench, where successful orchestration behavior is typically concentrated in the first 4–6 steps.
>
> ---
>
> ### **Response to Weakness 2, 5**
>
> **W2: Theoretical Rationale**. Our entropy dynamics model is not purely phenomenological. It is grounded in task logic and derived from 2 components:
>
> 1. momentum-based convergence, inspired by optimization theory
> 2. information entropy diffusion, motivated by information-theoretic uncertainty accumulation.
>
> The full derivation is provided in Appendix B.
>
> **W5: Constrained System**. We agree that many realistic systems combine auto-orchestration with fixed workflow fragments. This is compatible with our framework because each step in our model represents an orchestrator-level decision, rather than a single agent action. As shown in our case study, one step may involve multiple agents internally. Such structured fragments can therefore be treated as a single execution node without affecting the validity of our entropy measure.

---

> > ### Author Rebuttal · Reviewer_fYUP · 2026-04-04
> >
> > I am not yet fully convinced that the entropy dynamics formulation reflects a sufficiently principled model of the orchestrator’s decision process, and questions around how well the findings translate beyond synthesized environments to real-world behavior still remain. Given these concerns, I will increase my score to a weak reject.

---

> > > ### Author Response · Authors · 2026-04-07
> > >
> > > We sincerely thank you for the constructive discussion. To address your concern about findings that translate beyond synthesized environments to real-world behavior, we conducted an additional experiment in a real-world environment. It contains:
> > > - the real-world web environment on April 5, 2026, including live online queries, real-time data fetching, and time-sensitive information retrieval;
> > > - Magentic-One system [1], a representative real-world multi-agent system developed by Microsoft;
> > > - GPT-4o as the common base model and GAIA [2] as the task input.
> > >
> > > The experimental results are as follows:
> > >
> > > | Metric                | A_Task | γ    | ω    | β    |
> > > |-----------------------|--------|------|------|------|
> > > | Real-world   | 6.42   | 1.14 | 1.03 | 0.68 |
> > > | IWG                   | 3.16   | 1.56 | 1.72 | 0.47 |
> > >
> > > Successful parameter fitting of the mean-field entropy dynamics equation in real-world environments shows that it can be used to analyze the behaviors of real-world MAS deployments. A bigger A_Task value may be due to noises (such as network latency and API instability) in real-world environments triggering the context management mechanism of MAS, which converts these challenges into filtering noise and producing intermediate summaries. This also explains the decrease in γ and ω: more effort is spent on context consolidation and memory maintenance, leaving less budget for aggressive decomposition and exploration. Therefore, the main impact of realistic environments is not simply more context, but harder information management and execution under noise. Besides, there are similar fitting values of real-world environments and IWG, which indicates that our IWG approximates the real-world environment.
> > >
> > > Overall, the above results demonstrate the applicability of our framework to real-world environments.
> > >
> > > [1] Microsoft Research AI Frontiers. "Magentic-One: A Generalist Multi-Agent System for Solving Complex Tasks." arXiv preprint arXiv:2411.04468.
> > >
> > > [2] Mialon, Grégoire, et al. "GAIA: a benchmark for General AI Assistants." ICLR. 2024.

---

### Decision · Program_Chairs · 2026-04-30

**Decision:**

Accept (regular)

**Comment:**

We have carefully considered all the available evidence (the reviews, author responses, discussions, additional comments, and the paper).

While the paper is on a relevant and timely topic, and it makes a sound contribution, following the rebuttals and discussions, the reviewers remained divided on its readiness (in terms of overall methodology, assumptions, and theoretical and empirical results versus the claims made) for acceptance at ICML. The Weak Reject recommendation reviewers indicated very high levels of confidence, and the Strong Accept and Accept reviewers 3 confidence level.

Our recommendation was reached considering: (i) the two strong accept recommendations and the two Weak Reject recommendations; (ii) the extensive positive comments on the paper's methodology and contributions; (iii) the thorough and effective engagement of the authors via rebuttals and discussions; and (iv) the timeliness and relevance of the paper's topic to the ICML community.

We encourage the authors to carefully consider all the informative and constructive reviewers' comments and suggestions, and integrate their responses during the rebuttals and discussions in the final version of the paper.